# Learning probability distributions of sensory inputs with Monte Carlo predictive coding

**Gaspard Oliviers**[1]*, **Rafal Bogacz**[1], **Alexander Meulemans**[2]

**1** MRC Brain Network Dynamics Unit, Nuffield Department of Clinical Neurosciences, University of Oxford, Oxford, United Kingdom, **2** Department of Computer Science, ETH Zurich, Zürich, Switzerland

* gaspard.oliviers@bndu.ox.ac.uk

## Abstract

It has been suggested that the brain employs probabilistic generative models to optimally interpret sensory information. This hypothesis has been formalised in distinct frameworks, focusing on explaining separate phenomena. On one hand, classic predictive coding theory proposed how the probabilistic models can be learned by networks of neurons employing local synaptic plasticity. On the other hand, neural sampling theories have demonstrated how stochastic dynamics enable neural circuits to represent the posterior distributions of latent states of the environment. These frameworks were brought together by variational filtering that introduced neural sampling to predictive coding. Here, we consider a variant of variational filtering for static inputs, to which we refer as Monte Carlo predictive coding (MCPC). We demonstrate that the integration of predictive coding with neural sampling results in a neural network that learns precise generative models using local computation and plasticity. The neural dynamics of MCPC infer the posterior distributions of the latent states in the presence of sensory inputs, and can generate likely inputs in their absence. Furthermore, MCPC captures the experimental observations on the variability of neural activity during perceptual tasks. By combining predictive coding and neural sampling, MCPC can account for both sets of neural data that previously had been explained by these individual frameworks.

## Author summary

Understanding how the brain interprets its sensory information is fundamental to neuroscience. It is suggested that the brain processes information by updating models of the environment that exist inside the brain. These models make educated guesses about the world, relying on the noisy information received through our senses. However, translating this conceptual framework into a concrete, biological theory is challenging. Several proposed theories explain specific aspects of brain function or dynamics. For instance, predictive coding describes the organization of the brain which is important for understanding how the brain infers and learns. Other theories, such as neural sampling, use random changes in the brain's activity to explain how the brain interprets its sensory inputs. However, these theories remain separate, each explaining only certain brain

**Data Availability Statement:** The MNIST dataset is publicly available at https://yann.lecun.com/exdb/mnist/. The codebase with all models and experiments can be found at the following link:

https://github.com/gaspardol/
MonteCarloPredictiveCoding.git.

**Funding:** This work has been supported by the Medical Research Council UK (https://www.ukri.org/councils/mrc/) (grant MC_UU_00003/1 awarded to R.B). The funders had no role in study design, data collection and analysis, decision to publish, or preparation of the manuscript.

**Competing interests:** I have read the journal's policy and the authors of this manuscript have the following competing interests: R.B. is a shareholder in Fractile, Ltd., which designs artificial intelligence accelerator hardware. The remaining authors declare no competing interests.

functions. Our research introduces a theory that combines predictive coding and neural sampling into a unified framework for understanding brain learning and information processing. This model mirrors the brain's organization, information processing capabilities using local computations, and learning using local plasticity. It also accounts for experimentally observed characteristics of the brain's activity, while relying on minimal assumptions. Overall, our model offers a more comprehensive understanding of the brain's learning capabilities, relevant to both neuroscience and machine learning.

## 1 Introduction

The Bayesian brain hypothesis states that the brain learns and updates probabilistic generative models of its sensory inputs. By learning efficient generative models, the brain establishes the causal relationship between environmental states and sensory inputs [1–3]. The brain also mitigates the effect of sensory noise through generative models by optimally integrating prior knowledge with new sensory data. Several studies have successfully employed probabilistic generative models to explain behavior [4–9], and interpret neural activity [10–14].

To elucidate how the brain represents generative models, we seek a neural network capable of learning generative models, while adhering to the brain's intrinsic characteristics. These characteristics include (i) the brain's ability to infer posterior distributions of environmental states given sensory inputs [4–7, 15], (ii) its proficiency in constructing efficient and generalisable generative models using hierarchical neural networks [16], and (iii) its reliance on localized computation and plasticity within these networks [17, 18].

Multiple models implementing the Bayesian brain principle have been proposed that capture some of the above characteristics of the brain. Below we review two categories of models that focus on describing learning of probabilistic models and representing the posterior probabilities of the latent states respectively.

An influential theory describing how the cortex learns the generative models is predictive coding. It hypothesises that the brain learns the generative models by minimising the error between actual sensory inputs and the sensory inputs predicted by its model [19–21]. To support this theory, several neural networks have been proposed to illustrate how the brain might implement predictive coding [21–23]. These networks are hierarchically structured and are local in computation and plasticity. Moreover, since the time predictive coding was first proposed in neuroscience to explain retinal processing [24], it has evolved into a comprehensive framework for understanding attention [25], a range of neurological disorders [26], and various neural phenomena [19, 27, 28]. However, predictive coding has demonstrated a limited learning performance for generative tasks [29]. Recent work extended predictive coding to improve its learning performance using lateral inhibition and sparse priors [29, 30], however the resulting neural network is unable to infer posterior distributions or generate sensory samples. In addition to predictive coding, other models have been proposed to describe learning of probabilistic models in the brain. For example, a recent study has employed generative adversarial networks to explain delusions observed in some mental disorders [31]. However, no biologically plausible neural implementation of the adversarial objective function has been identified.

On the other hand, a wide range of neural sampling models have also been proposed that infer the posterior distributions using Monte Carlo sampling methods [12, 32–37]. In these models, the fluctuations of neural activity over time sample the probability distributions the brain is trying to infer. Some studies show that neural variability in the brain exhibits

characteristics consistent with neural sampling processes [38–40]. Despite this, present neural sampling models lack learning capabilities, local learning rules, or depth in their neural architectures. Recent work incorporated neural sampling into a sparse coding model that can learn generative models with local plasticity [34]. However, the sparse coding model does not include a hierarchical architecture that can support learning of complex generative models.

Here, we bring together the above work on predictive coding and neural sampling by proposing Monte Carlo predictive coding (MCPC). MCPC follows the approach of variational filtering [41] by integrating neural sampling into predictive coding, albeit in a simplified variant that disregards the dynamics of stimuli. This simplification offers two advantages: (i) the simplified model only differs from the predictive coding framework proposed by Rao and Ballard [19] in additional noise into its inference dynamics, allowing the application of recent advancements in neural implementations [42] and modeling of a broad range of brain learning tasks [43–46] to MCPC; (ii) it facilitates a more thorough evaluation of inference, generation, and learning performance of the model given that most benchmarks and metrics are designed for static inputs.

Monte Carlo predictive coding presents a biologically plausible neural implementation of generative learning in the brain. It infers full posteriors and learns hierarchical generative models by relying solely on local computation and plasticity. Furthermore, MCPC can generate sensory inputs using local neural dynamics, and its neural activity captures the variability in cortical activity during perceptual tasks. MCPC effectively learns generative models that generalise from data and robustly learns the data (co)variance structure across noise types and intensities as well. Overall, MCPC offers a comprehensive theoretical framework for understanding neural computation and capturing key characteristics of cortical activity.

## 2 Results

This section presents MCPC, and it is organized into subsections discussing the following properties of the model:

1. MCPC utilizes neural networks with local computation and plasticity to learn hierarchical generative models.

2. MCPC's neural dynamics infer full posterior distributions of latent variables in the presence of sensory inputs.

3. MCPC's neural dynamics sample from the learned generative model in the absence of sensory inputs.

4. MCPC learns efficient generative models that generalise from sensory data.

5. MCPC captures the variability of neural activity observed in perceptual experiments.

6. MCPC achieves robust learning of data (co)variance across noise types and intensities.

Throughout our experiments, we consider two tasks: learning a simple probability distribution of Gaussian sensory data, and learning a more complex distribution of handwritten digit images from the MNIST dataset [47]. We compare the properties of our model to predictive coding following the formulation by Rao and Ballard [19] and Bogacz [21] that we refer to with PC. This is because MCPC's neural dynamics are closely related to this implementation of predictive coding and the performance of this formulation of predictive coding has also been characterised in a variety of tasks (it achieves performances similar to backpropagation in supervised machine learning tasks [43], and superior to backpropagation in tasks more similar to those faced by biological organisms [44]). A comparison between MCPC and other

formulations of predictive coding that use techniques such as divisive input modulation [23], free-energy minimisation combined with the Laplace approximation [22], or variational filtering [41] is provided in the discussion.

## 2.1 MCPC implementation with local computation and plasticity

To describe MCPC, we will first define a hierarchical generative model MCPC assumes, next present its inference and learning algorithm, and then show how it can be implemented through local computation and plasticity.

MCPC learns a hierarchical Gaussian model of sensory input $y$ with latent variables $x$. The latent variables are organized into $L$ layers in this model. We denote the activity of sensory neurons by $x_0$, and when the sensory input is present, they are fixed to it, i.e., $x_0 = y$. Sensory input $y$ is predicted by the first layer $x_1$ while variables $x_l$ in layer $l$ are predicted by the layer above. The resulting joint distribution over sensory inputs and latent variables is given by:

$$p(y, x; \theta) = \prod_{l=0}^{L-1} \mathcal{N}(x_l; W_l f(x_{l+1}), \sigma^2 I) \mathcal{N}(x_L; \mu, \sigma^2 I) \tag{1}$$

where $x$ denotes the latent states $x_1$ to $x_L$, parameters $\theta$ comprise weights $W_l$ and the prior mean $\mu$ describing the mean activity in the top layer, $f$ stands for an activation function, $I$ represents an identity matrix, and $\sigma^2$ denotes a scalar variance. A simple example of such probabilistic model is illustrated in Fig 1a, and it includes one sensory input and one latent state. Such model could for instance be used by an organism to infer the size of a food item based on observed light intensity [21]. We will use this model throughout the paper to provide intuition before considering more complex models.

MCPC learns a hierarchical Gaussian model by iterating over two steps that descend the negative joint log-likelihood

$$F = -\ln p(y, x; \theta) = \frac{1}{2} \sum_{l=0}^{L-1} \frac{\| x_l - W_l \cdot f(x_{l+1}) \|^2}{\sigma^2} + \frac{1}{2} \frac{\| x_L - \mu \|^2}{\sigma^2} \tag{2}$$

In the first step, MCPC leverages Markov chain Monte Carlo techniques to approximate the full posterior distribution by using the following Langevin dynamics [48]:

$$\frac{\partial x_l(t)}{\partial t} = -\nabla_{x_l} F + n_l(t) \tag{3}$$

Thus we modify the latent variables to reduce $F$, but additionally add a zero-mean noise $n_l(t)$ (in the next subsection we will show explicitly that such dynamics lead to $x_l$ sampling

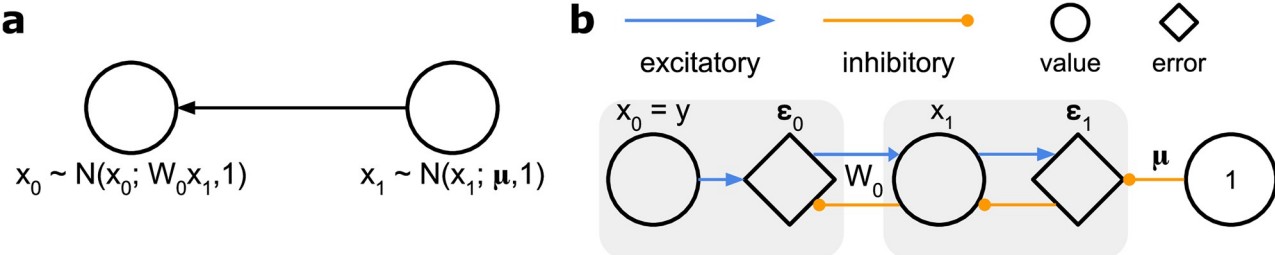

**Fig 1. Example of a probabilistic model and its corresponding neural implementation for MCPC. a,** Linear Gaussian model with one sensory input and one latent state. **b,** The neural implementation of MCPC using local synaptic connections for this model.

from its posterior distribution). The noise needs to be uncorrelated over time, i.e. with covariance $\mathbb{E}\left[n_l(t)n_l(t')^\top\right] = 2\sigma_n^2\delta(t - t')I$, where $\delta$ is the Dirac delta function, $\sigma_n^2$ is the noise variance and $I$ the identity matrix. The noise variance $\sigma_n^2$ is set to one unless otherwise stated.

Evaluating the gradient in Eq 3, we see below that these neural dynamics give rise to prediction errors $\epsilon_l$ encoding the mismatch between the predicted latent state $W_l\, f(x_{l+1})$ and the inferred latent state $x_l$.

$$\frac{\partial x_l(t)}{\partial t} = -\epsilon_l + f'(x_l)W_{l-1}^\top\epsilon_{l-1} + \ n_l(t), \tag{4}$$

$$\epsilon_l = x_l - W_l f(x_{l+1}); \quad \epsilon_L = x_L - \mu \tag{5}$$

In the second step, MCPC uses the noisy neural activities to update its parameters as follows:

$$\Delta W_l \quad \propto -\int\limits_{t_0}^{t_0+T} \nabla_{W_l}F\mathrm{d}t = \int\limits_{t_0}^{t_0+T} \epsilon_l(t)f(x_{l+1}(t))^\top\mathrm{d}t \tag{6}$$

$$\Delta\mu \quad \propto -\int\limits_{t_0}^{t_0+T} \nabla_\mu F\mathrm{d}t = \int\limits_{t_0}^{t_0+T} \epsilon_L(t)\mathrm{d}t \tag{7}$$

with $t_0$ the time point where the noisy dynamics have converged to their steady-state distribution, and $T$ is large to ensure that the dynamics sample from the whole steady-state distribution. Repeating these two steps enables inference of latent variables and learning of model parameters.

The above algorithm has a direct implementation in a neural network. Such a network has two classes of neurons: value neurons encoding latent states and error neurons encoding prediction errors. The weights of synaptic connections in such a network encode the parameters of the generative model. This is illustrated in Fig 1b through a simple network implementing probabilistic inference in the model from Fig 1a.

The neural network of MCPC relies on local computation. All neurons perform computations solely based on the activity of their input neurons and the synaptic weights related to these inputs. Specifically, the rate of change of value neurons in Eq 4 depends on their own activity, the activity of the error neurons connected with them, the weights of these connections, and local noise. Similarly, the activity of error neurons in Eq 5 can be computed using the activity of connected value neurons and corresponding synaptic weights.

The network also exhibits local plasticity. Synaptic plasticity in MCPC (Eqs 6 and 7) relies exclusively on the product of the activity of pre-synaptic and post-synaptic neurons. The integral in MCPC's synaptic plasticity can also be approximated using local plasticity. This could be achieved by continuously updating synaptic weights with a large time constant.

The neural dynamics and parameters updates of MCPC prescribe the same local neural circuits as existing implementations of predictive coding [21, 49], with the addition of a noise term. Hence, MCPC shares the focus of predictive coding on minimizing prediction errors. The additional noise term does, however, lead to significant benefits as discussed below.

## 2.2 MCPC infers posterior distributions

Here we show that MCPC's neural activity infers full posterior distributions of latent variables in the presence of sensory inputs. We prove that MCPC's neural activity samples the posterior $p(x|y; \theta)$ at its steady state for an input $y$. Moreover, we confirm that MCPC's neural activity approximates the posterior in the linear model of Fig 1a and in a model trained on MNIST digits.

Proposition 1 demonstrates that the neural activity $x$ prescribed by MCPC samples from the posterior $p(x|y; \theta)$ over latent states $x$ when the dynamics in Eq 3 have converged.

**Proposition 1** *The posterior $p(x|y; \theta)$ is the steady-state distribution $p^{ss}(x)$ of the inference dynamics of MCPC:*

$$p^{ss}(x) = \frac{e^{-F}}{Z} = \frac{e^{\ln p(y,x;\theta)}}{Z} = \frac{p(y;\theta)}{Z} p(x|y;\theta) = p(x|y;\theta) \tag{8}$$

*where Z is the partition function.*

The proof is given in the transformations in Eq 8, which we now explain. It follows from a classical result in statistical physics that the steady-state distribution $p^{ss}(x)$ of a variable $x$ described by the Langevin equation $\frac{\partial x}{\partial t} = -\nabla_x F + n(t)$ is given by $p^{ss}(x) = e^{-F}/Z$ when the variance of the noise $\sigma_n^2 = 1$ [50]. The Langevin dynamics of MCPC minimise the negative joint log-likelihood $F = -\ln p(y, x; \theta)$. The distribution $p^{ss}(x)$ can therefore be rewritten as $p(y, x; \theta)/Z$. Employing the conditional probability formula allows $p^{ss}(x)$ to be subsequently expressed as $p(y; \theta)p(x|y; \theta)/Z$. Given that the distribution $p(y; \theta)$ remains constant for a particular stimulus $y$, $\frac{p(y;\theta)}{Z}$ forms the partition function of the posterior $p(x|y; \theta)$. However, the posterior distribution integrates to one, $\int p(x|y; \theta)dx = 1$. This implies that $\frac{p(y;\theta)}{Z}$ equals one and that the steady-state distribution $p^{ss}(x)$ effectively simplifies to the posterior distribution $p(x|y; \theta)$. This result for MCPC is analogous to the use of Langevin dynamics for posterior inference in other models [12, 34, 51].

To verify this property, we validate that MCPC samples from the posterior distribution within the simple model from Fig 1a, which is tractable. Fig 2a illustrates the activity of latent state $x_1$ of this model during inference under a constant input for both MCPC and PC. While the activity converges to a single value for PC, activity for MCPC fluctuates around this value representing the uncertainty in its inference. Fig 2b displays a histogram of the latent state's activity over time throughout the MCPC inference. The inference of PC at its convergence point is also illustrated, as well as the posterior distribution $p(x_1|y; \theta)$ for the specified input. MCPC's latent state activity accurately samples the posterior of the linear model. In contrast, PC's inference converges to the mode of the posterior. This result confirms that MCPC samples from the posterior $p(x|y; \theta)$, whereas PC infers the Maximum a-posteriori (MAP) estimate.

Next, we visually confirm that MCPC infers latent states correctly in a non-linear model with three latent layers trained on MNIST digits. Visualising the latent states during inference shows that both MCPC and PC infer the correct digit when provided with a full-digit image (Fig 2c). However, when prompted with an ambiguous masked-digit image, MCPC identifies different possible interpretations, while PC only infers one possible interpretation (Fig 2d). This result indicates that MCPC approximates the posterior distribution more accurately than PC. The visualisations are obtained by employing a linear classifier to interpret the latent states. This classifier decodes the latent state $x_L$ and generates a probability distribution over the ten-digit categories. This distribution can then be visualised by mapping it onto ten evenly spaced unit vectors within a circle [52](see Methods section 4.2.2 for details). The activity of

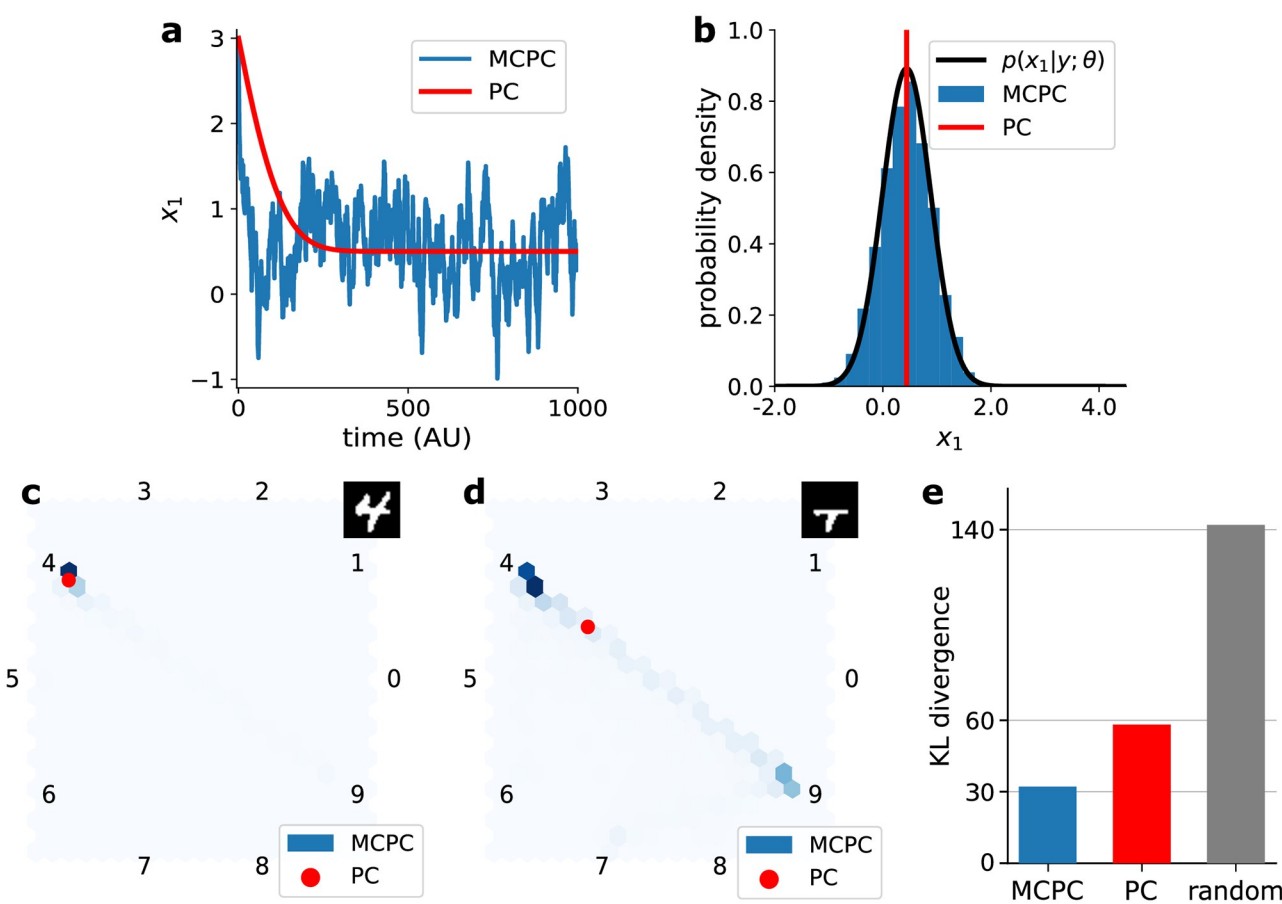

**Fig 2. Neural activity of MCPC infers posterior distributions in the presence of inputs. a,b,** Latent state activity $x_1$ of MCPC and PC in the linear model shown in Fig 1a with parameters $\{W_0 = 2, \mu = 0.5\}$ and input $y = 1$. **c,d,** Latent state activity of MCPC and PC in a model trained on MNIST with a digit image and a half-masked digit image (see top-right) as input. Plots (**b**), (**c**), and (**d**) show a histogram of MCPC's activity over 10,000 timesteps and PC's activity at converges. **e,** KL divergence between the digit class distribution inferred by an ideal ResNet-9 observer and the class distribution decoded from the latent state $x_L$ inferred by MCPC and PC for masked digit images. The KL divergence for shuffled distributions is also provided. Animation of the MCPC's latent activity in plots b to d can be found in S1 Video, S2 Video and S3 Video.

latent layer $x_L$ is visualised here. However, similar results are observed for all latent layers, as detailed in S1 Fig.

Finally, we show quantitatively that MCPC indeed approximates the posterior better than a MAP estimate in a non-linear model trained on MNIST. Fig 2e shows the Kullback–Leibler (KL) divergence between the posterior across digit classes inferred by a ResNet-9-based ideal observer and the distributions inferred by MCPC, and PC for half-masked images. The KL divergence for a random baseline obtained with shuffled distributions is also shown. This figure shows that the KL divergence between the distributions inferred by the ideal observer and by MCPC is smaller than the one for PC and for the baseline. The lower KL divergence confirms that MCPC's inferred latent states capture the posterior distribution more accurately than PC's MAP estimate. In this experiment, ResNet-9 is a classifier that achieves over 99% classification accuracy on MNIST [53]. The probability distributions across digit classes of MCPC and PC inferences are obtained with the linear classifier used for interpreting the latent states. Moreover, the random baseline is calculated by averaging the KL divergence between the inferences of the ideal observer and the shuffled distributions inferred by MCPC and by PC.

## 2.3 MCPC samples from its generative model in the absence of sensory inputs

Here we show that in the absence of sensory inputs, MCPC spontaneously samples from its learned generative model of sensory inputs. We prove that the activity of the unclamped input neurons sample from probability distributions of sensory inputs learned by MCPC. Experiments confirm that the unclamped neural activity generates sensory inputs learned by MCPC in the simple model of Fig 1a and in a model trained on MNIST.

To model a scenario in which no sensory input is provided, instead of clamping the input neurons $x_0$ to a sensory stimulus $y$, we let these neurons follow similar Langevin dynamics as all other neurons:

$$\frac{\partial x_0}{\partial t} = -\nabla_{x_0} F + n_0(t) = -\epsilon_0 + n_0(t). \tag{9}$$

Proposition 2 shows that when input neurons are not fixed to sensory stimuli, MCPC spontaneously samples from the learned probability distribution of sensory inputs. In this proposition, we demonstrate that the steady state of MCPC's unclamped activity is equal to the marginal likelihood $p(x_0; \theta)$.

**Proposition 2** *The marginal likelihood $p(x_0; \theta)$ is the steady-state distribution $p^{ss}(x_0)$ of the Langevin dynamics given in* Eq 9:

$$p^{ss}(x_0) \quad = \int \frac{e^{-F}}{Z} dx = \int \frac{e^{\ln p(y=x_0, x; \theta)}}{Z} dx = p(x_0; \theta) \int \frac{p(x|x_0; \theta)}{Z} dx \tag{10}$$

$$= p(x_0; \theta) \tag{11}$$

The proof of proposition 2 is similar to that of proposition 1. The steady-state distributions of the neural activity in MCPC in the absence of an input $p^{ss}(x_0, x)$ is given by $e^{-F}/Z$. This is a consequence of the Langevin dynamics of MCPC minimizing the negative joint log-likelihood $F$ while subjected to a noise variable with variance $\sigma_n^2 = 1$. This distribution can be marginalised over the latent states $x = [x_1, \ldots, x_L]$ to find the steady-state distribution of the sensory input neurons $p^{ss}(x_0)$. The joint log-likelihood $F$ equals $-\ln p(y, x; \theta)$, where $y = x_0$ when input neurons are unclamped. The expression for $p^{ss}(x_0)$ is therefore reformulated as $\int e^{lnp(y=x_0, x; \theta)} / Z dx$. This expression can be rewritten as $p(x_0) \int p(x|x_0)/Z dx$. Given that the expression $\int p(x|x_0)/Z dx$ remains constant for a specific activity $x_0$, this expression forms the partition function of the marginal likelihood $p(x_0; \theta)$. However, the marginal likelihood integrates to one, $\int p(x_0; \theta) dx = 1$. This implies that the partition function $\int p(x|x_0)/Z dx$ equals one and that the steady-state distribution $p^{ss}(x_0)$ effectively simplifies to $p(x_0; \theta)$.

Fig 3a and 3b experimentally confirm that MCPC generates accurate samples of the generative distribution in the absence of sensory inputs. Fig 3a demonstrates this for the linear model by showing that the activity of the unclamped input neuron matches the model's generative distribution $p(x_0; \theta)$. Similarly, Fig 3b illustrates that the unclamped neural activity of a deep non-linear model trained on MNIST produce activity patterns that resemble the digit images used in training.

## 2.4 MCPC learns efficient and generalisable generative models

We show here that MCPC learns precise generative models of sensory data. We demonstrate the precise learning of MCPC by first proving that MCPC is guaranteed to converge to a local

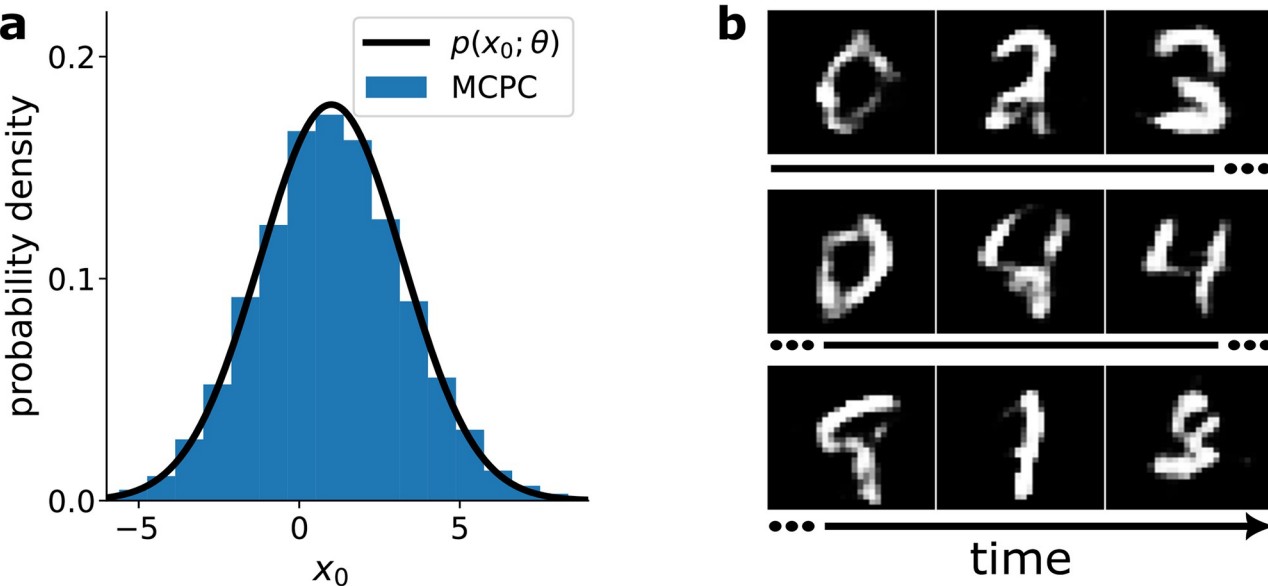

**Fig 3. Neural activity of MCPC samples its generative model in the absence of inputs. a**, Histogram of the MCPC activity of the unclamped input neuron $x_0$ in the linear model given in Fig 1a with parameters $\{W_0 = 2, \mu = 0.5\}$ obtained over 10,000 timesteps. **b**, Activity patterns generated by a model trained on MNIST displayed for time points separated by 3,000 timesteps. The samples of the MNIST-trained model display the probability of a sensory neuron being equal to one. Animation of the MCPC's unclamped input activity can be found in S4 Video and S5 Video.

optimum of the marginal likelihood $p(y; \theta)$. Afterwards, we experimentally confirm that MCPC learns efficient generative models of Gaussian sensory data and handwritten digit images. In the process, MCPC outperforms PC and approaches the performance of Deep Latent Gaussian models (DLGMs) on the digit learning task. DLGMs are the standard machine learning approach for training hierarchical Gaussian models (Eq 1) using backpropagation [54], and are therefore used as a benchmark. Lastly, we show that MCPC learns hierarchies of abstractions indicating its capacity for generalisation.

MCPC learns locally optimal generative models of sensory data by implementing the Monte Carlo expectation-maximization algorithm (see proposition 3). This algorithm guarantees that the model parameters converge to a local optimum of the marginal likelihood $p(y; \theta)$ when given enough sampling time during inference [55].

**Proposition 3** *MCPC implements the Monte Carlo expectation-maximization algorithm by iterating over:*

*1. E-step: MCPC's inference, $x(t)$, approximates the posterior distributions using an MCMC method for a given input y*

$$x(t) \sim p(x|y; \theta),$$

*2. M-step: MCPC's parameter update maximizes the Monte Carlo expectation of joint log-likelihood*

$$\Delta \theta \propto \nabla_\theta \int_{t_0}^{t_0+T} \ln p(y, x(t); \theta) \mathrm{d}t \propto \nabla_\theta \mathbb{E}_{p(x(t))} \{ \ln p(y, x(t); \theta) \}.$$

Proposition 3 relies on proving that MCPC's inference samples the posterior distribution for a given input and proving that MCPC's parameter updates maximize the Monte Carlo expectation of the joint log-likelihood. Proposition 1 shows that MCPC's inference samples

the posterior distribution. This provides half of the proof for proposition 3. The second part of the proof can be shown by identifying that the expressions $F$ in Eqs 6 and 7 equal the negative joint log-likelihood. This allows MCPC's parameter updates to be rewritten as $\int_{t_0}^{t_0+T} \nabla_\theta \ln p(y, x(t); \theta) \mathrm{d}t$. The partial derivatives can be taken out of the integrals to obtain the parameter update $\nabla_\theta \int_{t_0}^{t_0+T} \ln p(y, x(t); \theta) \mathrm{d}t$. In this expression, $\int_{t_0}^{t_0+T} \ln p(y, x(t); \theta) \mathrm{d}t$ is the Monte Carlo expectation of the joint log-likelihood. Consequently, MCPC parameter updates maximise the Monte Carlo expectation of joint log-likelihood.

Additionally, we show that MCPC learns the distribution of Gaussian sensory data with the linear model of Fig 1a. Fig 4a illustrates the distribution learned by MCPC after 375 parameter updates. This distribution models the Gaussian data distribution used for training. We obtain the samples of the distribution learned by MCPC using ancestral sampling. In a hierarchical Gaussian model, ancestral sampling consists of first sampling the top latent layer $x_L$ from its Gaussian distribution $\mathcal{N}(x_L; \mu I)$. Each layer is then sampled sequentially using the conditional Gaussian distribution $\mathcal{N}(x_l; W_l f(x_{l+1}), I)$. Fig 4b verifies that MCPC learns an accurate model of Gaussian data for different model initialisations. This figure demonstrates that each parameter trajectory converges to the parameters for ideal data modeling. This convergence can also be validated analytically by first calculating the curves where the parameter update for the

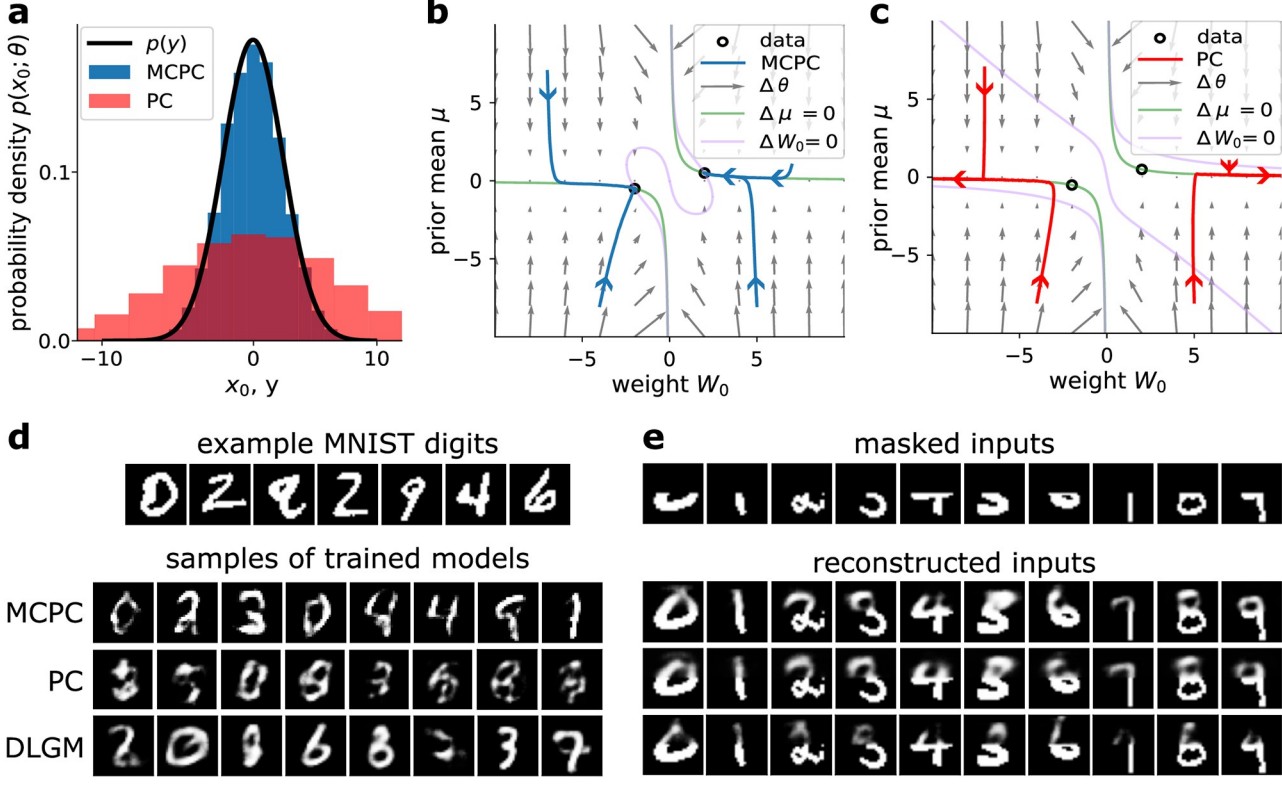

**Fig 4. MCPC learns efficient generative models of sensory inputs. a**, Distributions learned by MCPC and PC in the linear model given in Fig 1a after 375 parameter updates. **b,c**, Evolution of the weight $W_0$ and prior mean $\mu$ parameter of the linear model during training with MCPC (**b**) and PC (**c**). The optimal model parameter values are marked as hollow dots. The vector field shows the expected gradient flow of the parameters. The additional curves reveal nullclines where the parameter update for the weight or the prior mean parameter equals zero (see S1 Appendix for derivations). **d**, Comparison between samples obtained from models trained with MCPC and PC on MNIST, as well as from a DLGM trained on MNIST. The samples are obtained by ancestrally sampling the models for PC and the DLGM and by sampling the spontaneous neural activity for MCPC. **e**, Comparison between masked images reconstructed by MCPC, PC, and a DLGM. We reconstruct the images by obtaining a Maximum a-posteriori estimate of the missing pixel values.

weight or the prior mean parameter equals zero (these curves are known as nullclines and shown in green and purple in Fig 4b). The intersection of these curves provides the equilibrium points for the parameter values. For MCPC, this intersection is located at the model parameters that perfectly capture the Gaussian data distribution (see S1 Appendix for full derivation).

In contrast to MCPC, PC learns a strikingly poor generative model of the Gaussian data as shown in Fig 4a. PC learns a Gaussian distribution with the correct mean but with an excessive variance. This high variance is caused by the diverges of PC's weight to $\pm\infty$ during training as shown in Fig 4c for different model initialisations. The variance of the model learned by PC equals $W_0^2 + 1$ (see Eq 19 in Methods). Consequently, the variance of PC's generative distribution grows toward infinity as training progresses, leading to a model that becomes increasingly inaccurate. PC's parameter $W_0$ diverges to $\pm\infty$ additionally validating the suboptimal learning performance of PC (see S1 Appendix). The underlying cause of this undesirable behavior of PC is that it uses the maximum a-posteriori estimate of $x_l$ in its parameter updates, instead of the full posterior distribution. This learning strategy is equivalent to variational expectation maximization [56], with a Dirac-delta as variational approximation to the true posterior. Crucially, the Dirac-delta ignores uncertainty and introduces an infinite entropy to the free-energy, causing the free-energy to become an arbitrarily loose bound on $\ln p(y; \theta)$ (refer to Olshausen [57] and S2 Appendix. for additional details). As a consequence, the marginal likelihood that needs to be optimised can not be evaluated, which in practise leads to PC's weights diverging. In contrast, MCPC implements the Monte Carlo expectation-maximisation algorithm that optimises the marginal log likelihood $\ln p(y; \theta)$ (Proposition 3). Interestingly, a range of other theories for learning in the brain [43, 58, 59] are based on a similar energy as in PC, posing the question of whether they suffer from similar failure modes as we uncover here for PC.

Next, we show that MCPC learns accurate hierarchical Gaussian models of MNIST handwritten digit images [47]. For this learning task, we consider non-linear models with three latent layers. We train these models using MCPC, PC, and the DLGM approach (refer to methods section 4.2.2 for details). Fig 4d presents samples generated from the trained models. The quality of samples generated from an MCPC-trained model approaches that of samples produced by a DLGM. However, the samples obtained from training with PC are of significantly poorer quality, even when we apply weight decay to mitigate PC's exploding variance. To quantify the difference in performance, we compute three metrics. First, we calculate the Fréchet inception distance (FID) of the generated samples which measures the similarity between generated data and actual data [60]. Second, we approximate the marginal log-likelihood of test data $\ln p(y_{eval}; \theta)$ using Monte Carlo sampling. This metric evaluates the generalization performance of a trained generative model. Third, we compute the mean squared error (MSE) associated with reconstructing masked digits as illustrated in Fig 4e. This assesses the ability to learn and retrieve associative memories [61]. Table 1 summarises the results and shows that a model trained with MCPC generates significantly better samples than PC and that it has better

**Table 1. Comparison of learning performance between MCPC, PC and DLGM.**

| Model | FID | $-\ln p(y_{eval})$ | Reconstruction MSE ($10^{-2}$) |
|-------|-----|--------------------|--------------------------------|
| PC | 115.2 ± 3.0 | 168.9 ± 0.2 | 8.73 ± 0.03 |
| MCPC | 60.6 ± 2.9 | 144.6 ± 0.7 | **8.29 ± 0.05** |
| DLGM | **45.4 ± 0.7** | **126.0 ± 0.3** | 12.04 ± 0.08 |

We report the FID, the marginal log-likelihood, and the reconstruction error on an MNIST evaluation set (the closer to zero the better for all metrics). We set in bold the best score across the models. Mean ± standard deviation computed over three seeds.

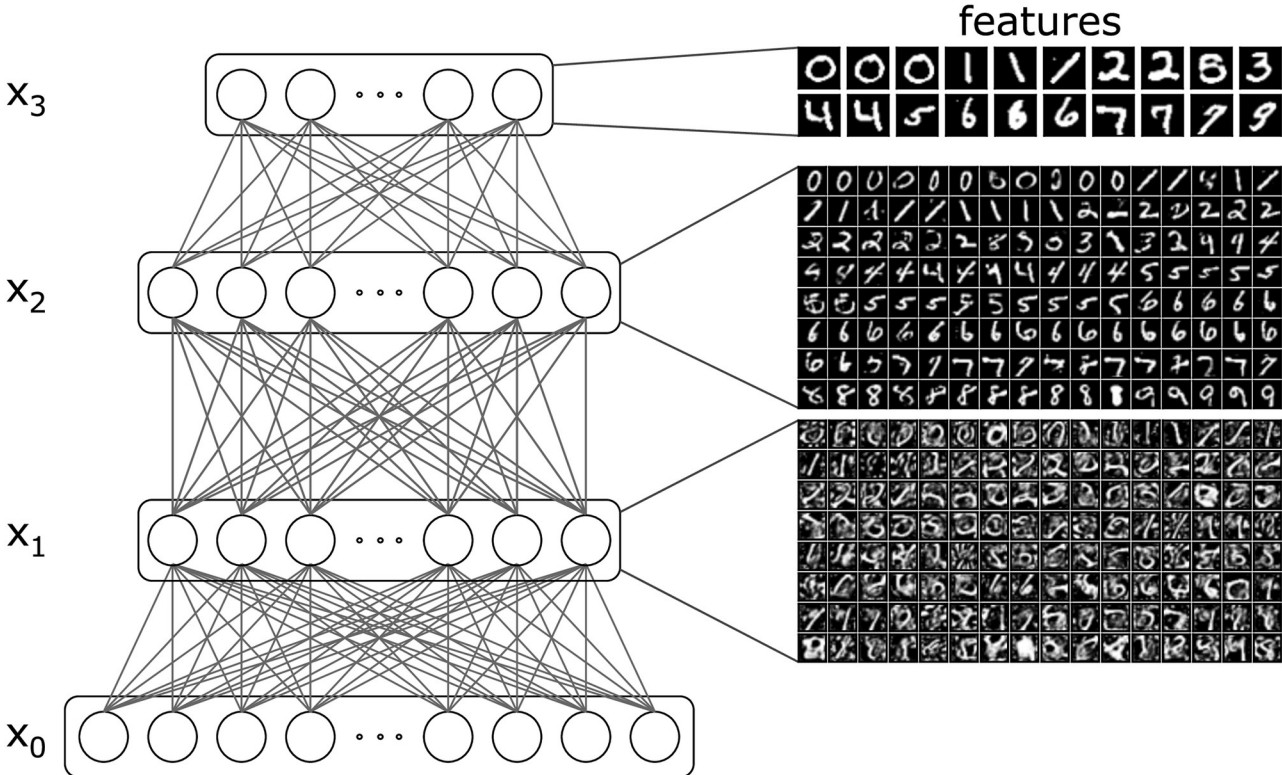

**Fig 5. MCPC learns hierarchies of abstractions.** Features of each neuron in an MCPC model trained on MNIST. The features are sorted using a ResNet-9 classifier for each layer.

generalization performance. Additionally, MCPC approaches the generative learning performance of DLGM. Table 1 also shows that MCPC can reconstruct masked digits as well as PC and that both perform significantly better than DLGMs.

Finally, we demonstrate that MCPC effectively learns hierarchical abstractions from data. Fig 5 illustrates the features learned by all latent neurons in a model trained using MCPC on MNIST. These features are extracted by setting all the neurons' activities in a layer to zero except for one neuron, which is set to a high activity level. The latent activity of that layer is then propagated forward through the model until the input layer is reached. This process is equivalent to finding the activity pattern that minimizes the negative joint log-likelihood conditioned on the manipulated layer of neurons. The features learned by the first latent layer, $x_1$, consist of 128 low-level features. The features learned by the second latent layer, $x_2$, include 128 digit representations that vary in orientation, shape, and style. The features learned by the final latent layer, $x_3$, consist of 20 digits encompassing most classes with minimal within-class variation. This progression indicates that the model learns increasingly abstract features from the input layer to the deepest latent layer. Furthermore, this demonstrates the model's ability to transition from representing pixel-level information in the lower layers to capturing semantic information in the higher layers, thereby showcasing its capacity for generalization.

### 2.5 MCPC captures the variability of cortical activity

MCPC captures the key characteristics of the variability of cortical activity during perceptual tasks that PC fails to capture. Specifically, MCPC accounts for the suppression of neural

variability at stimulus onset and the increase in similarity between spontaneous and evoked neural activities during development.

MCPC exhibits a decrease in temporal variability of neural activity at stimulus onset as observed in multiple electrophysiology studies [62–69]. These studies have shown that neural variability is smaller after stimulus onset than before stimulus onset. This finding holds when measured with intracellular or extracellular recordings and when an animal is task-engaged, awake, or anesthetized. Fig 6a illustrates the neural variability experimentally observed by Churchland et al. [65] and the neural variability of MCPC's latent states at stimulus onset. This figure shows that MCPC's neural activity captures the decrease in neural variability at stimulus onset for an MNIST-trained model. S3 Appendix provides additional proof that this observation generally holds for MCPC. This proof shows that the variability of MCPC's steady state activity before stimulus onset is in expectation larger than the variability after stimulus onset.

MCPC displays an increase in similarity between spontaneous and average evoked neural activities that is specific to natural scenes as observed during learning for ferrets [38]. Berkes et al. [38] recorded the spontaneous and average evoked neural activity in V1 of ferrets for natural stimuli, sinusoidal gratings, and random noise. They observed that, as development progressed, the spontaneous activity increasingly resembled the average activity evoked by natural stimuli. Additionally, this increase in similarity was not observed for the sinusoidal gratings and random noise. Fig 6b compares the similarity between spontaneous and evoked neural activities for natural stimuli, noise, and image gratings reported by Berkes et al. [38] and observed for MCPC in MNIST-trained models. MCPC displays an increase in similarity

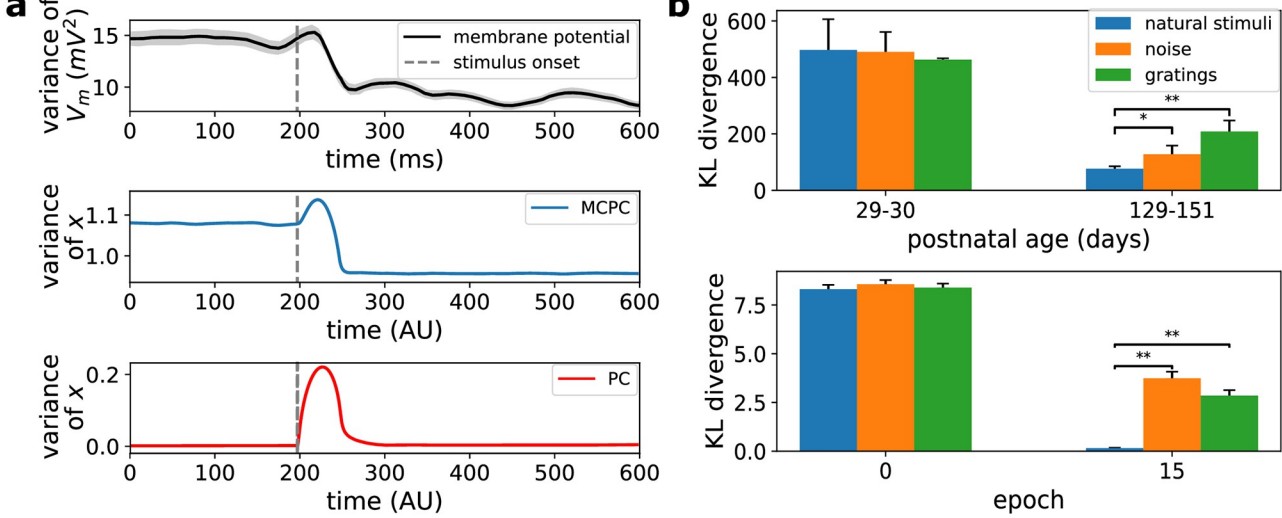

**Fig 6. MCPC captures two key features of cortical activity. a**, MCPC displays the decrease in neural variability at stimulus onset observed in the primary visual cortex (V1) of cats. The top plot recreates the neural quenching observed in the cortex (data re-plotted from figure 2c in Churchland et al. [65]). The middle and bottom plots show the mean temporal variability at stimulus onset of the latent state for MCPC and PC in a model trained on MNIST. Shaded regions give the s.e.m. Note that for MCPC and PC, these shaded regions are not visible due to their minimal magnitude. **b**, MCPC displays the similarity increase between spontaneous and evoked neural activities specific to natural stimuli observed in V1 of ferrets during development. The similarity is measured using the KL divergence between the distribution of spontaneous activity and the average distribution of evoked neural activities (the closer to zero the more similar). The average distribution is obtained for natural stimuli, noise stimuli, and gratings. The top plot recreates the similarity increase observed in awake ferrets (data re-plotted from figure 4a in Berkes et al. [38]). The bottom plot demonstrates a parallel increase in similarity specific to the training stimuli for MCPC. The MCPC model was trained on MNIST and evaluated using noise, image gratings, and MNIST digits (analogous to natural stimuli). In both plots, the error bars give the s.e.m. and $^*$ or $^{**}$ indicate $p < 0.05$ or $p < 0.01$ respectively for a one-tailed paired samples t-test based on the KL divergences obtained for $n = 10$ MCPC models with the same architecture but different initializations.

between spontaneous and evoked neural activities that is specific to the digit stimuli on which it was trained (that are analogous to natural scenes to which the visual systems of animals were exposed). Such an increase in the similarity between spontaneous activity and the average response to stimuli on which the model was trained holds for MCPC in general. This is because the steady-state distribution of MCPC's spontaneous neural activity becomes more similar to the average steady-state distribution of MCPC's evoked activity as MCPC's generative model improves (see S3 Appendix for proof). In our experiment, the similarity in neural activities is measured using the KL divergence for neurons in layer $x_1$ and natural images are MNIST images (see Methods section 4.2.2 for a detailed explanation of the experiment). The natural stimuli-specific similarity increase is present across the model's latent layers. However, as shown in S1 Fig, the KL divergence is only significantly smaller for natural stimuli in layer $x_1$.

Both the above characteristics of cortical activity are not reproduced by PC. The spontaneous and evoked neural activities of PC converge to constant neural activity without neural variability. Consequently, the temporal variability of individual neurons is not suppressed at stimulus onset in PC. Instead, the variability temporarily increases above zero at stimulus onset after which it returns to zero as illustrated in Fig 6a. Moreover, training does not enhance the similarity between the distributions of spontaneous and evoked activities in PC. The distribution of PC's spontaneous activity is a Dirac delta distribution as the activity has no variability. Similarly, the distribution of PC's average evoked activity is a Dirac mixture distribution. Consequently, the KL divergence between these two non-identical Dirac-based distributions is always infinite.

## 2.6 MCPC robustly learns the data (co)variance across noise types and intensities

We demonstrate that MCPC effectively learns the variance and covariance structure of data. MCPC is also flexible in accommodating any type of noise distribution and variance in its Langevin dynamics, thereby avoiding the introduction of biologically unrealistic assumptions in the model.

MCPC learns the variance and covariance structure of data for both Gaussian data and the MNIST dataset by capturing the data covariation in its model weights. Fig 7a and 7b demonstrate that the MCPC model in Fig 1a learns a distribution with the same variance as its Gaussian training data across a range of data variances, $\Sigma_{data}$. However, MCPC can not learn the data variance when the data variance is smaller than its layer variance $\sigma^2$ and the Langevin noise variance $\sigma_n^2$ equals one. For the considered linear model, the marginal likelihood $p(x_0; \theta)$ equals $\mathcal{N}(x_0; \mu W_0, \sigma^2(W_0^2 + 1))$. Therefore, the model's variance is directly encoded in the model's weight $W_0$ and the model can not learn the data distribution when the data variance is smaller than the layer variance $\sigma^2$. This result is experimentally confirmed in Fig 7c which shows that the learned weight $W_0$ approximately equals $\sqrt{\Sigma_{data}/\sigma^2 - 1}$ only for data variance larger than $\sigma^2$. MCPC also learns the covariation structure of data as shown in Fig 7d. This figure compares the absolute correlation between non-zero pixels in the MNIST dataset and image samples generated by MCPC, confirming that MCPC accurately learns the pixel correlations.

An additional noteworthy characteristic of the MCPC is its flexibility in accommodating any type of noise distribution and variance. The only requirements on the noise variable $n_l(t)$ in MCPC's dynamics are as follows: (1) the noise has a zero mean, (2) it is uncorrelated in time and across neurons, and (3) the variance of the noise needs to be constant over time. These requirements follow from the fluctuation-dissipation theorem in statistical mechanics that

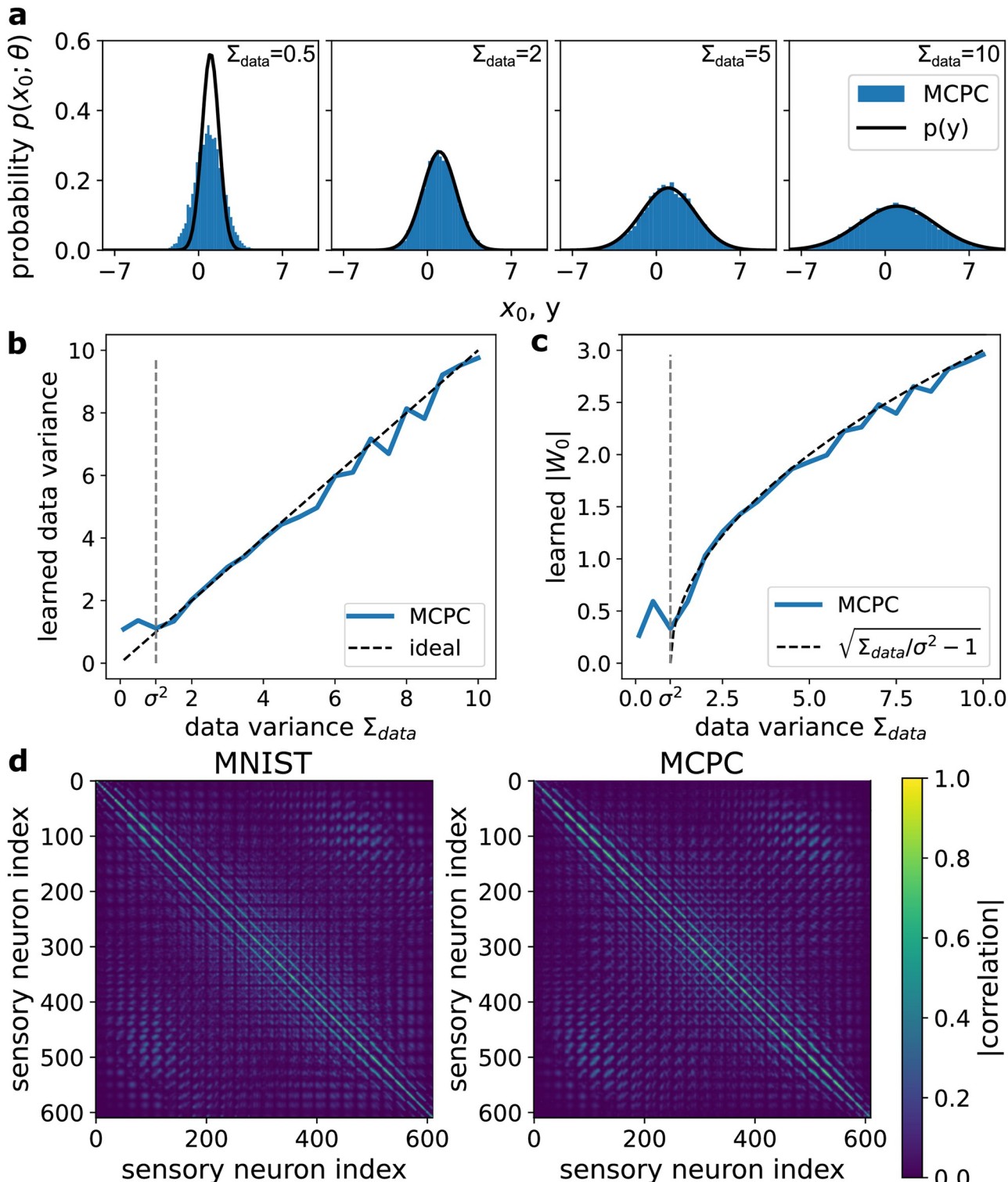

**Fig 7. (Co)variance learning in MCPC. a-c.** MCPC model of Fig 1a trained on Gaussian data for a range of training data variances. **a.** Comparison between data distribution and distribution generated by trained model. The distributions learned by MCPC are obtained using MCPC's spontaneous activity after 10,000 timesteps **b.** Variance of distribution generated by trained MCPC model for a range of training data variances. **c.** Absolute weight, $W_0$, of trained MCPC model for a range of training data variances. The ideal weight $W_0$ equals $\pm\sqrt{\Sigma_{data}/\sigma^2 - 1}$ where $\sigma^2 = 1$ in our experiment. Moreover, the vertical dashed line shown in (**b**) and (**c**) indicates where the variance of the Gaussian input layer of the MCPC model $\sigma^2 I$ becomes larger than the variance of the data distribution. **d.** Comparison between the correlation of pixels in the MNIST dataset and in 4000 images samples generated by an MNIST-trained MCPC model. Pixels that are always equal to zero in our MNIST evaluation set are excluded.

determines the first two moments of $n_l(t)$ (Eqs 12a and 12b) and the resulting steady-state distribution of the stochastic dynamics (Eq 13) where $\sigma_n^2$ scales the variance of the noise [50].

$$\mathbb{E}[n_l(t)] = 0 \tag{12a}$$

$$\mathbb{E}[n_l(t)n_l(t')\rangle] = 2\sigma_n^2 \; \delta(t - t')I \tag{12b}$$

$$p^{ss}(x) = \frac{e^{F/\sigma_n^2}}{Z} = \frac{e^{\sum_{l=0}^{L} \frac{1}{2}\|x_l - W_l f(x_{l+1})\|^2/(\sigma^2\sigma_n^2)}}{Z} \tag{13}$$

The fluctuation-dissipation theorem does not impose any specific constraints on the exact distribution of MCPC's noise [70]. This absence of assumption regarding the specific noise distribution ensures that MCPC does not hinge on potentially biologically implausible noise distributions.

The scalar variance of the noise, $\sigma_n^2$, is also not specified in the requirements of the fluctuation-dissipation theorem. As a result, it can be equal to values other than identity (default value used in experiments). However, altering $\sigma_n^2$ affects the variance of the layers in MCPC's generative model, as it changes the steady-state distribution of the Langevin dynamics. Specifically, this adjustment scales the variance of the generative layers to $\sigma^2\sigma_n^2 I$, as indicated in Eq 13. Therefore, $\sigma_n^2$ must remain constant over time to ensure consistency during learning and subsequent inferences.

To verify that MCPC learns the variance of data for non-identity scalar noise variances, we train the linear model from Fig 1a on Gaussian data with various levels of noise in its dynamics. Fig 8a and 8b show that MCPC accurately learns the data variance and distribution when the noise variance, $\sigma_n^2$, is below an upper limit. The learned distributions are generated using MCPC's unclamped neural activity while maintaining the level of noise used during training. For the model of Fig 1a, the marginal likelihood $p(x_0; \theta)$ equals $\mathcal{N}(x_0; \mu W_0, \sigma_n^2\sigma^2(W_0^2 + 1))$ for non-identity noise variances. The model weight, $W_0$, should therefore equal $\sqrt{(\Sigma_{data}/\sigma_n^2\sigma^2 - 1)}$ to capture the data variance and there should exist a learning limit at $\sigma_n^2 = \Sigma_{data}/\sigma^2$ above which no weight values exist to capture the data variance. Fig 8c confirms that the model weight changes according to $\sqrt{(\Sigma_{data}/\sigma_n^2 - 1)}$ to capture the data variance. Moreover, a learning limit exists at $\sigma_n^2 = \Sigma_{data}/\sigma^2$ above which MCPC maximally reduces the model's variance by setting the weight $W_0$ close to zero.

## 3 Discussion

This work establishes how the brain could learn probability distributions of sensory inputs by relying solely on local computations and plasticity. We propose Monte Carlo predictive coding, a neural model that learns probability distributions of sensory inputs using a hierarchical neural network with local computation and plasticity. MCPC introduces neural sampling to predictive coding using Langevin dynamics which enables: (i) the inference of full posteriors, (ii) the sampling of learned sensory inputs analogous to the brain imagining sensory stimuli, (iii) learning accurate generative models of sensory inputs, (iv) an ability to explain the variability of cortical activity and (v) learning data variance robustly across noise types and intensities.

### 3.1 Benefits from computational abilities of MCPC

The identified neural dynamics of MCPC infer posterior distributions and generate data samples, and these abilities would provide great benefits to organisms supporting them. On one

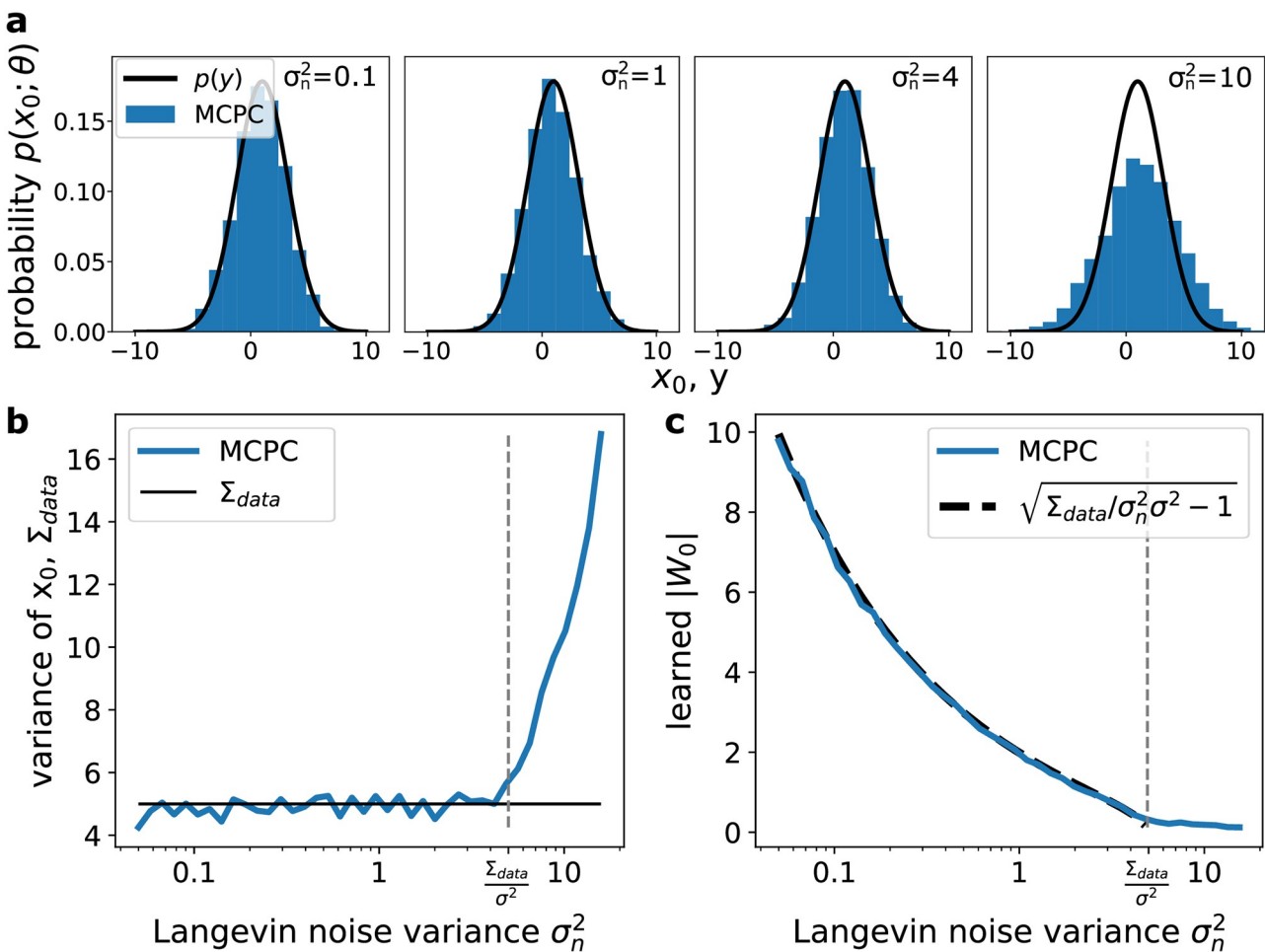

**Fig 8. MCPC is compatible with a range of noise levels. a**, Distributions learned by MCPC for the linear model shown in Fig 1a when trained on Gaussian data with four different levels of noise. **b**, Comparison between the variance of the data distribution and the variance of the distribution learned by MCPC with a range of noise levels. **c**, Comparison between the weight parameter $W_0$ learned by MCPC and the ideal weight for different levels of noise. The ideal weight parameter is given by $\pm\sqrt{\Sigma_{data}/\sigma_n^2\sigma^2 - 1}$ which can be found by comparing the marginal likelihood of the model to the data distribution as shown in section 4.2.1. The Gaussian data used for training in all panels has a variance of five. The distributions learned by MCPC in (**a**) and (**b**) are obtained using MCPC's spontaneous activity over 10,000 timesteps after training while maintaining the level of noise used during training. Moreover, the vertical dashed line shown in (**b**) and (**c**) indicates where the variance of the Gaussian input layer of the MCPC model $\sigma_n^2\sigma^2 I$ becomes larger than the variance of the data distribution. In these experiments, $\Sigma_{data} = 5$, $\mu_{data} = 1$ and $\sigma^2 = 1$.

hand, the ability of MCPC to infer posterior distributions reflects the brain's ability to infer statistically optimal representations of the environment. Such representations are key for survival through optimal perception [71] and decision-making [72]. On the other hand, our model's ability to generate samples from learned sensory inputs is essential for offline replay. Cognitive functions that rely on offline replay include memory consolidation [73], planning of future actions [74], visual understanding [75], predictions [76], and decision-making [77]. Taken together, the neural activity of MCPC provides the basis upon which a wide array of other brain functions depend. This implies that MCPC might be useful not only for understanding generative learning, but also for unraveling the brain functions that potentially depend on its neural activity patterns.

### 3.2 Unified theory of cortical computation

MCPC integrates the strengths of predictive coding and neural sampling providing a unified theory of cortical computation.

MCPC, as a form of predictive coding, utilizes prediction error minimization for inference and learning in a hierarchical model. This alignment with predictive coding enables the application of its potential cortical microcircuit implementations [78] and its implementation using dendritic errors [42] to MCPC. Additionally, MCPC can be applied to various learning tasks, similar to PC. For example, PC shows promising results in various classical tasks such as supervised learning, associative learning, representational learning, and reinforcement learning [30, 44, 79]. We expect MCPC to surpass PC in these tasks, owing to its enhanced inference dynamics that more closely approximate posterior distributions.

Concurrently, MCPC embodies neural sampling by employing neural dynamics to sample posterior distributions. Neural sampling was first proposed by Hoyer and Hyvärinen [10]. Since then, different implementations of sampling-based computations by the brain have been proposed [12, 32–37, 40]. These models have offered valuable insights that could be applied to MCPC. For instance, the sampling efficiency of MCPC could be improved through the use of excitatory and inhibitory recurrent networks, as suggested by Hennequin et al. [80]. Ultimately, MCPC opens new possibilities for a more comprehensive understanding of cortical computation and of the interplay between prediction-based learning and stochastic sampling mechanisms within the brain.

As a theory of cortical computation, MCPC can provide a account for a broad spectrum of cortical phenomena. This theory could bridge the explanatory scopes of both predictive coding and neural sampling. Predictive coding has played a pivotal role in providing a unified framework for explaining perception and attention [25]. It simultaneously offers insights into a range of neurological disorders such as schizophrenia, epilepsy, post-traumatic stress disorder, and chronic pain [26]. Predictive coding has also explained diverse neural phenomena ranging from retinal information encoding [27], alpha oscillations [28], and non-classical receptive fields [19]. Neural sampling has provided significant insights in explaining dynamic features of cortical activity. These features include the stimulus-dependence of neural variability [65, 81] and oscillations in the gamma band [82], strong transients at stimulus onset [83], and the spatiotemporal dynamics of bi-stable perception [6, 7]. By integrating predictive coding with neural sampling, MCPC is poised to offer a comprehensive model capable of bridging the explanatory scopes of both predictive coding and neural sampling.

### 3.3 Relationship to implementations of predictive coding

Here, we compare MCPC to several formulations of inference and learning using predictive coding.

Our experiment compares MCPC to predictive coding as described by Rao and Ballard [19] and Bogacz [21]. This predictive coding model uses a Dirac delta function to approximate the true posterior during inference. It relies on local computation and plasticity but is limited to MAP inference and does not support effective learning, unlike MCPC.

PC/BC-DIM is another version of predictive coding that aligns with biased competition theories of cortical function [84, 85]. PC/BC-DIM uses divisive input modulation [86] to update error and prediction neuron activations. This method supports local computation and offers faster inference, but in contrast to MCPC, it does not infer the uncertainty of its inferences.

Another predictive coding implementation by Friston and Kiebel [22] uses normal distributions to approximate the true posterior during inference. This approach utilises the same neural dynamics as Rao and Ballard [19], initially performing MAP inference. Later, it uses the Laplace approximation to approximate the posterior as a normal distribution around the inferred mode. While the MAP inference remains local, estimating the variance of the Laplace approximation in multivariate models involves non-local computation. Compared to MCPC, it allows for faster inference. However, this method approximates the posterior rather than fully sampling it, which can cause problems in complex models with multimodal posteriors.

Finally, variational filtering represents another predictive coding implementation that utilizes Langevin dynamics for inference, similar to MCPC. This approach operates within generalized coordinates, which is beneficial for learning dynamic latent variables and temporal structures in data. MCPC can be considered a zero-order version of variational filtering, where sensory input remains static over time. However, in its current form, MCPC cannot learn dynamic inputs. To extend MCPC to accommodate time-varying inputs, a scheme similar to the recently developed temporal predictive coding [45, 87] could be employed, which uses an additional set of weights to predict future latent states from past latent states.

## 3.4 Relationship to other models introducing Langevin dynamics to brain-inspired generative models

Several brain-inspired generative models using Langevin dynamics have been proposed, and here we discuss their similarities and differences from MCPC.

Langevin dynamics were initially proposed as a sampling strategy for posterior inference that is neurally implementable [12, 36]. This research paved the way for other models that elucidate diverse facets of perception and cortical functions using neural sampling [32, 40, 80]. For instance, the two experimental observations of neural variability captured by MCPC, as demonstrated in our work in Fig 6, have been explained using neural sampling [81]. Nevertheless, the proposed neural sampling models are either devoid of learning capabilities or rely on non-local plasticity mechanisms for weight adjustment.

Langevin dynamics have also been applied in sparse coding models for posterior inference [34]. These models leverage local Langevin dynamics for inference and employ local plasticity rules specific to sparse coding for learning. When trained on patches of natural images, these models successfully learn simple-cell receptive fields. Unlike MCPC, these sparse coding models do not possess hierarchical structures. Additionally, their learning capabilities have only been evaluated on relatively simple datasets, such as oriented bars.

Several machine learning studies have shown that generative models with Langevin dynamics learn accurate generative models of complex machine learning tasks [51, 88]. The studies show that the model with Langevin dynamics can outperform Variational Autoencoder and Generative adversarial networks on datasets such as MNIST, CIFAR-10, and CelebA. These studies confirm that models with Langevin dynamics can learn accurate generative models. However, in contrast to MCPC, these studies considered models that learn using non-local plasticity.

Since our initial presentation of MCPC [89], subsequent research [90, 91] has further validated that generative models employing Langevin dynamics learn precise generative models on complex tasks. Zahid et al. [90] also proposed the use of Langevin dynamics in predictive coding. However, their investigation focused on biologically implausible models with a singular latent layer, trained via backpropagation, diverging from MCPC's approach. On the other hand, Dong and Wu [91] incorporated Langevin dynamics into generative models that leverage local computation and plasticity, showcasing capabilities for posterior inference and data

generation using local neural dynamics akin to MCPC. Dong and Wu [91] employ exponential-family energy-based models, which differ from the hierarchical Gaussian models used in predictive coding. As a result, their proposed models are less directly linked to predictive coding than MCPC.

## 3.5 Experimental prediction

The core prediction of MCPC posits that the brain concurrently performs predictive error computations and sampling processes. Experimentation to substantiate MCPC would therefore involve detecting simultaneous prediction errors and neural sampling. According to predictive coding theories, prediction errors can be measured in the activity of error neurons, as discussed in this paper, or in the activity of dendrites [42]. Notably, this activity intensifies in response to unanticipated sensory inputs. Additionally, a measurable signature of sampling is a change in neural variability of value-encoding neurons as a result of a change in uncertainty associated with sensory inputs. An experimental approach to test MCPC's prediction could, therefore, involve training animals to classify visual stimuli. Following their training, the experiment would measure the neural responses in the animals' primary visual cortex when they are shown ambiguous and unambiguous stimuli. MCPC predicts that: (i) Neurons or dendrites that encode prediction errors will exhibit greater activity in response to ambiguous stimuli compared to non-ambiguous stimuli, and (ii) value-encoding neurons involved in sampling will display increased variability when processing ambiguous stimuli as opposed to unambiguous stimuli. An observed increase in both error-encoding activity and variability in value neurons in response to ambiguous stimuli, compared to unambiguous ones, would suggest the brain's use of principles similar to those in MCPC for generative learning.

## 3.6 Limitations and future work

**3.6.1 Extending MCPC to learn precision weighted prediction errors.**   While MCPC effectively learns data variance and covariance over a wide range of data parameters, it encounters learning limitations with narrow data distributions relative to its layer variance. To address this issue, the model could be extended to parameterize and learn the precision matrices of the Gaussian layers within MCPC's generative model, where the precision matrix is the inverse of the covariance matrix. This extension would be particularly significant for the predictive coding field, as existing literature emphasizes the importance of minimizing precision-weighted prediction errors for both computation and neuropathology [25, 26]. Various schemes for predictive coding models have been proposed to learn precision matrices through local computations [21, 46]. These schemes could be applied to MCPC by appropriately modifying the negative log-likelihood function *F*.

**3.6.2 Improving the sampling speed of Langevin dynamics.**   Despite the promising results in this study, sampling using MCPC's Langevin dynamics requires long inference times [48]. The inference duration could be significantly shortened by relying on advancements in neuroscience and machine learning. For instance, Hennequin et al. [80] showed how the cortex could increase the sampling efficiency of neural circuits using excitatory and inhibitory recurrent networks. Adding higher-order terms to the Langevin dynamics, such as momentum, also dramatically improves convergence speed [92]. Additionally, Ma et al. [93] proposed a general framework for improving the sampling efficiency of Langevin-based sampling. By applying these principles to MCPC, the sampling speed is expected to increase, reaching a value that resembles the fast sampling of the brain [1]. Importantly, the learning performance of MCPC is then anticipated to improve, as the inferences will capture the posterior more effectively.

The convergence speed of MCPC's Langevin dynamics also increases with the number of latent dimensions. In our experiments, the model trained on MNIST (over 200 dimensions) requires significantly more inference steps to reach a steady state than the model trained on Gaussian data (1 dimension), as shown in S2 Fig. Scaling MCPC to large models might therefore be limited by the number of inference steps required to reach steady state which may exceed practical limits.

**3.6.3 Mapping MCPC's noise to sources of noise in the brain.** Currently, mapping the noise variable in MCPC's neural dynamics to distinct noise sources in the brain remains a challenge. Cortical circuits have various forms of stochasticity that could support the random dynamics of MCPC [94]. However, the constraints on the noise variable within the dynamics of MCPC are notably minimal. MCPC does not mandate that the noise follow a particular distribution, nor does it specify a required noise level. Consequently, predicting which types of noise in the brain could facilitate the stochastic dynamics of MCPC proves challenging. To establish a more direct link between MCPC's noise and its potential physiological origins, a spiking implementation of MCPC could be identified. Rethinking MCPC using spiking neural networks, as done in the spiking models of predictive coding [95], might add constraints on the location and type of noise needed. These constraints could create a clear connection to the physiological sources of noise.

# 4 Methods

## 4.1 Models

In this paper, we compare three methods for learning hierarchical Gaussian models: Monte Carlo predictive coding, predictive coding, and backpropagation in a deep latent Gaussian model.

**4.1.1 Monte Carlo predictive coding.** Algorithm 1 shows the complete implementation of MCPC used for all the simulations in the paper. This algorithm is a discrete-time equivalent of MCPC where the dynamics of MCPC given in in Eq 3 are discretized using the Euler–Maruyama method. The algorithm contains a MAP inference before the MCPC inference to shorten MCPC's mixing time during inference. This additional MAP inference is, however, not always beneficial as discussed in S3 Fig.

Moreover, the algorithm learns using mini-batches of data. Inference is performed independently for each element in a mini-batch and parameters are updated using the sum of the parameter updates across the mini-batch.

**Algorithm 1:** Monte Carlo predictive coding (MCPC)

```
Require: L layers, activities x₀ to x_L, noise variance σ²ₙ, weights W₀
          to W_{L-1}, joint log-likelihood F, dataset {y_p}^P_{p=1} with P mini-
          batches of B elements, number of epochs E, Euler step h, num-
          ber of Euler steps K, number of mixing steps M, number of sam-
          pling steps S, and learning rate α.
for e = 1 to E do
  for p =1 to P do
    // Independent inference for each sample in batch
    x_{0,b} ← y_{p,b}, 1 ≤ b ≤ B
    x_{l,b} ← n,    n ∼ 𝒩(0,I), 1 ≤ l ≤ L and 1 ≤ b ≤ B
    // MAP inference for faster steady-state
    for k = 1 to K do
      x_{l,b} ← x_{l,b} - h (∂F_b/∂x_{l,b}),   1 ≤ l ≤ L and 1 ≤ b ≤ B
    // MCPC inference
    for i = 1 to M + S do
      x_{l,b} ← x_{l,b} - h (∂F_b/∂x_{l,b}) + √(2h) n_{l,b},
```

$$n_{l,b} \sim \mathcal{N}(0, \sigma_n^2 I), 1 \le l \le L \text{ and } 1 \le b \le B$$

$x(i)_b \leftarrow x_b \quad 1 \le b \le B$

`// Sum of parameter updates for batch`

$W_l \leftarrow W_l - \frac{\alpha}{S} \sum_{i=M+1}^{M+S} \sum_b^B \frac{\partial F(y_{p,b}, x(i)_b; \{W, \mu\})}{\partial W_l}, \quad 1 \le l \le L-1$

$\mu \leftarrow \mu - \frac{\alpha}{S} \sum_{i=M+1}^{M+S} \sum_b^B \frac{\partial F(y_{p,b}, x(i)_b; \{W, \mu\})}{\partial \mu}$

**Algorithm 2:** Predictive coding (PC)

**Require:** $L$ layers, activities $x_0$ to $x_L$, weights $W_0$ to $W_{L-1}$, joint log-likelihood $F_{pc}$, dataset $\{y_p\}_{p=1}^P$ with $P$ mini-batches of $B$ elements, number of epochs $E$, Euler step $h$, number of Euler steps $K$ and learning rate $\alpha$.

**for** $e$ = 1 **to** $E$ **do**

 **for** $p$ = 1 **to** $P$ **do**

 `// Independent inference for each sample in batch`

 $x_{0,b} \leftarrow y_{p,b}, \quad 1 \le b \le B$

 $x_{l,b} \leftarrow n, \quad n \sim \mathcal{N}(0, I), 1 \le l \le L \text{ and } 1 \le b \le B$

 **for** $k$ = 1 **to** $K$ **do**

 $x_{l,b} \leftarrow x_{l,b} - h \frac{\partial F_{pc,b}}{\partial x_{l,b}}, \quad 1 \le l \le L \text{ and } 1 \le b \le B$

 `// Sum of parameter updates for batch`

 $W_l \leftarrow W_l - \alpha \sum_b^B \frac{\partial F_{pc,b}}{\partial W_l}, \quad 1 \le l \le L-1$

 $\mu \leftarrow \mu - \alpha \sum_b^B \frac{\partial F_{pc,b}}{\partial \mu}$

**4.1.2 Predictive coding.** We briefly review the predictive coding framework and its implementation used in this paper. Following the formulation of predictive coding by Bogacz [21] which we refer to with PC, predictive coding learns a hierarchical Gaussian model. The model is learned by iterating over two steps that minimise the joint log-likelihood $F_{pc} = -\ln p(y, x; \theta) = \frac{1}{\sigma^2} \sum_{l=0}^L \| x_l - W_l \cdot f(x_{l+1}) \|^2$ where $x_0$ is clamped to an observation $y$. First, PC uses neural dynamics that follow the gradient flow on $F_{pc}$ to infer the Maximum a-posteriori estimate of the latent states conditioned on the observation:

$$\frac{\partial x_l(t)}{\partial t} = -\nabla_{x_l} F_{pc} = -\epsilon_l + f'(x_l) W_{l-1}^\top \epsilon_{l-1} \tag{14}$$

$$\epsilon_l = \frac{1}{\sigma^2}(x_l - W_l f(x_{l+1})); \quad \epsilon_L = \frac{1}{\sigma^2}(x_L - \mu) \tag{15}$$

Second, PC updates the parameters with a gradient step on $F_{pc}$, evaluated on the converged MAP estimate $x^*$ with error $\epsilon^*$:

$$\Delta W_l \propto -\nabla_{W_l} F_{pc} = \epsilon_l^* f(x_{l+1}^*)^\top; \quad \Delta \mu \propto -\nabla_\mu F_{pc} = \epsilon_L^*. \tag{16}$$

These computations can be implemented in the same neural network with local computation and plasticity as MCPC. This is because PC only differs from MCPC through an additional noise term in the neural dynamics and an integral in the weight updates. Algorithm 2 shows the complete implementation of PC used for all the simulations in the paper. This algorithm is a discrete-time equivalent of PC where the dynamics of PC given in Eq 14 are discretized using Euler's method which consists of taking small discrete steps in the derivative direction.

**4.1.3 Deep latent Gaussian models.** We implement DLGMs as a benchmark model because they are the standard machine learning model for learning hierarchical Gaussian models. DLGMs were first proposed by Rezende et al. [54] and they consist of two main components: a generative model and an inference model. Each of these models is represented by separate neural networks. The generative model is responsible for generating

samples, while the inference model approximates the posterior distribution over the latent variables given the observed data. To train this model, we utilize the reparameterization trick [54, 96]. This technique allows for the backpropagation of gradients through stochastic nodes, enabling efficient and accurate gradient-based optimization to learn the parameters of both the generative and inference models. However, DLGMs are not a biologically plausible model of generative learning in the brain. One major shortcoming is that the plasticity mechanisms used by DLGMs are not local. This study uses a modified version of the DLGMs implementation by Zhuo [97]. The implementation is modified so that the inference network learns a rank 1 approximation of the covariance matrices of the posterior. This reduces the number of parameters of the inference network without significantly affecting the learning performance of DLGMs. To ensure a fair comparison with PC and MCPC, we employ DLGMs with generative models possessing a parameter count equivalent to that of PC and MCPC. Furthermore, the inference networks of DLGMs are constrained to maintain a parameter count equal to their generative counterparts.

## 4.2 Learning tasks

Throughout the paper two generative learning tasks are studied: a Gaussian learning task and a handwritten digit image learning task.

**4.2.1 Gaussian learning task.** In this task, the data has a Gaussian distribution that can be learned by the model in Fig 1a. This model is tractable, facilitating a comparison of MCPC's steady state inference with the marginal likelihood $p(y; \theta)$ and the posterior distribution $p(x|y; \theta)$. The model's tractability also enables a direct comparison of the optimal parameters to the parameters learned by MCPC and PC. Eqs 17 and 18 provide the data distribution and the model used for this task.

$$p(y) = \mathcal{N}(y; \mu_{data} = 1, \Sigma_{data} = 5) \tag{17}$$

$$p(y, x; \theta) = \mathcal{N}(y; W_0 x_1, \sigma^2 \sigma_n^2 I) \mathcal{N}(x_1; \mu, \sigma^2 \sigma_n^2 I) \tag{18}$$

The marginal likelihood and the posterior distributions are given in Eqs 19 and 20.

$$p(y; \theta) = \mathcal{N}(y; W_0 \mu, \sigma^2 \sigma_n^2 (W_0^2 + I)) \tag{19}$$

$$p(x|y; \theta) = \frac{p(y, x; \theta)}{p(y; \theta)} = \frac{\mathcal{N}(y; W_0 x_1, \sigma^2 \sigma_n^2 I) \mathcal{N}(x_1; \mu, \sigma^2 \sigma_n^2 I)}{\mathcal{N}(y; W_0 \mu, \sigma^2 \sigma_n^2 (W_0^2 + I))} \tag{20}$$

Unless otherwise stated, both $\sigma^2$ and $\sigma_n^2$ are set to one. To evaluate steady-state neural activity with and without inputs, we employ 10,000 inference steps for MCPC and 2,000 steps for PC. The optimal parameter values, $\{W_{0,opt} = \pm\sqrt{\Sigma_{data}/\sigma^2 \sigma_n^2 - 1}, \mu_{opt} = \pm\mu_{data}/\sqrt{\Sigma_{data}/\sigma^2 \sigma_n^2 - 1}\}$, are identified by comparing the marginal likelihood to the data distribution. We train an MCPC and a PC model on this task using the parameters in Table 2.

**4.2.2 MNIST learning task.** In this task, the dataset comprises 28x28 binary images representing handwritten digits across ten categories. The model architecture and training parameters used for this task are determined using a hyperparameter search. Moreover, the model has been adapted to have a Bernoulli input layer. In contrast to the Gaussian learning task, the model is intractable due to its hierarchical structure and non-linear activation functions, necessary for accurate learning. Consequently, direct assessment of inferences and model parameters is not feasible. Instead, we visualize the neural activity of MCPC and PC with and without

**Table 2. Parameters of MCPC, PC, and DLGMs.**

| | Gaussian | MNIST |
|---|---|---|
| MCPC Inference | | |
| optimizer | SGD | SGD |
| lr | 0.02 | {0.003, 0.01, 0.03, 0.1, 0.3} |
| mixing steps $T$ | 150 | 50 |
| sampling steps $M$ | 1 | 100 |
| PC Inference | | |
| optimizer | Adam | Adam |
| lr | 0.02 | {0.03, 0.1, 0.3, 0.7} |
| max_steps | 150 | 250 |
| Architecture | | |
| input dimension | 1 | 748 |
| number of latent layers $L$ | 1 | 3 |
| dimension of $x_1$ to $x_{L-1}$ | - | {128, 256, 360} |
| dimension of $x_L$ | 1 | {10, 15, 20, 25, 30} |
| activation function | linear | {ReLU, tanh} |
| layer variance $\sigma^2$ | 1 | 1 |
| noise variance $\sigma_n^2$ | 1 | 1 |
| Learning | | |
| optimiser | Adam | Adam |
| lr | 0.02 | {0.001, 0.003, 0.01, 0.03} |
| decay | 0 | {0, 0.01, 0.1, 1} |
| num_epochs | 75 | 50 |
| batch_size | 256 | {64, 128, 256} |

The parameters are given for the linear task and the hyperparameter search space for the MNIST task. The parameters under MCPC Inference are only used for MCPC inference. The parameters under PC Inference are used for PC inference and the MAP inference in algorithm 1 implementing MCPC. The parameters under Architecture and Learning are shared by MCPC, PC, and DLGMs.

inputs, we compare the neural activity of MCPC and PC with inputs to the inferences of an artificial ideal observer, and we measure the learning performance of the models using three metrics.

**Model parameters.** The model architecture and training parameters used for this task are determined using a hyperparameter search summarised in Table 2. The dataset includes 60,000 training images and 10,000 testing images, of which 6,000 images are used for hyper-parameter tuning and 4,000 for evaluation. S4 Appendix compile the search results.

**Bernoulli input layer.** The model used for this task has been adapted to have a Bernoulli input layer. For binary images, the model's input layer is transformed into a multivariate Bernoulli distribution. This modification yields the joint log-likelihood $F_{Bernoulli} = -[y^\top \ln(s(W_0 f(x_1))) + (1 - y^\top) \ln(1 - s(W_0 f(x_1)))] + \sum_{l=1}^{L} (x_l - W_l \cdot f(x_{l+1}))^2$ where $s$ is a sigmoid function. This change does not compromise the biological plausibility of MCPC because it only introduces an additional non-linearity in the inference dynamics of $x_1$ and the

parameter update for $W_0$. This modification is shown below:

$$\frac{\partial x_1}{\partial t} = -\nabla_{x_1} F_{Bernoulli} + n_0(t)$$
$$= -\epsilon_1 + W_0^\top \epsilon_0 + n_0(t) \qquad \text{with } \epsilon_0 = [y - s(W_0 f(x_1))]$$
$$\Delta W_0 \propto \int_{t_0}^{t_0+T} -\nabla_{W_0} F_{Bernoulli} \mathrm{d}t$$
$$\propto \int_{t_0}^{t_0+T} \epsilon_0(t) f(x_1(t))^\top \mathrm{d}t.$$

**Visualisation of MCPC's and PC's neural activity.** We visualize MCPC's and PC's neural activity to assess the inference with and without inputs. We record and display the input neurons' activity over time to visualize inferences without inputs. For the model trained on MNIST, the input Bernoulli sensory layer is discrete and cannot utilize Langevin dynamics. Therefore, we record the neural activity of the model excluding the Bernoulli sensory layer and display the input to the Bernoulli layer predicted by the first latent layers. Mathematically, this is represented as $s(W_0 f(x_1))$. To visualize the neural activity with inputs, we use a linear classifier that decodes the digit class distribution from the latent state $x_L$. The classifier is trained on full images of the training data to transform MAP inferences of the latent state $x_L$ to the corresponding digit classes. The digit classes are then assigned coordinates using a convex combination of 10 evenly spaced points on a unit circle [52], resulting in a two-dimensional visualization. In Fig 2, we visualize the inference for a full image part of the evaluation data and a partially masked version of the same image. For both visualizations, we use 10,000 inference steps for MCPC and 2,000 for PC.

**Quantification of neural activity of MCPC and PC with inputs.** We compare the neural activity inferred by MCPC and PC with inputs to the posterior inferred by an artificial ideal observer. This comparison quantifies how well MCPC and PC approximate the posterior distribution. The artificial ideal observer is a ResNet-9 classifier. We employ the same linear classifier as used for the visualizations to decode a digit class distribution from the neural activity of MCPC and PC. The decoded class distribution can then be compared to the digit class distribution inferred by the ideal observed. For PC, the digit class distribution is obtained by decoding the inferred latent state $x_L$ at convergence. For MCPC, this distribution is obtained by decoding the fluctuating latent state $x_L$ at steady state and averaging the decoded distributions across MCPC samples. MCPC and PC are compared to the ideal observer by computing the KL divergence between the digit class distributions on the MNIST evaluation set with the top half of the images masked. For the random baseline, we compute the Kullback-Leibler divergence between the posterior distribution inferred by the ideal observer and the distributions inferred by both MCPC and PC, after these have been randomly shuffled. This shuffling results in the distributions inferred by PC and MCPC being associated with random inputs.

**Performance metrics.** We assess the learning accuracy of MCPC and PC using three metrics. Firstly, the Frechet Inception Distance (FID) evaluates the quality and diversity of generated images [60]. The FID is computed by comparing the evaluation images with 5000 generated images using a public FID implementation [98]. Secondly, we approximate the marginal log-likelihood for the evaluation images to assess a model's generalization performance. The marginal log-likelihood is approximated using the following Monte Carlo estimate from

5000 latent state samples:

$$-\ln p(y_{eval}; \theta) \quad = -\ln \prod_{i=1}^{4000} \int p(y_{i,eval}, x; \theta)dx$$

$$\approx -\sum_{i=1}^{4000} \ln \sum_{s=1}^{5000} p(y_{i,eval}, x_s; \theta), \quad x_s \sim p(x; \theta)$$

The samples to compute both the FID and the marginal log-likelihood are obtained using ancestral sampling. Finally, the reconstruction MSE measures the error between images and the images reconstructed by a model when inputted with the bottom half of the images. We calculate the error as the mean squared error between the top half of the original and reconstructed images for the MNIST evaluation set. The reconstructed images are obtained by performing MAP inferences of the missing image pixel values.

**Visualising model features.** We visualize the model features of all latent neurons for a model trained on MNIST with MCPC with hyperparameters that optimize the model's FID. The feature represented by a neuron is determined by setting the activity of that neuron to 10 while keeping all other neurons in the same layer at zero activity. This approach is due to the use of ReLU non-linearities in the model, which disregard any activities less than or equal to zero. The chosen activity level of 10 for the analyzed neuron is based on experimental findings showing that activities of 10 or higher produce the same generated pattern. The neuron's feature is obtained by propagating the modified layer activity forward through the model's non-linearities and parameters until it reaches the input layer. For example, after setting the activity in layer $x_2$ the feature would be calculated as $s(W_0 f(W_1 f(x_2)))$.

**Pixel Correlation.** We visualize and compare the absolute values of the Pearson correlation coefficients between pixels in MNIST images from our evaluation set and those generated by a trained MCPC model optimized for the best FID score. Pixels that are consistently zero in the MNIST evaluation set are excluded, as the correlation coefficient is undefined for these pixels. We compute the correlation coefficients for 4000 image samples generated using ancestral sampling from the MCPC model trained on MNIST.

### 4.3.1 Measuring cortical-like properties of MCPC's neural activity

We measure two properties of MCPC models: the neural variability at stimulus onset, and the similarity between spontaneous and average evoked activity during training.

**4.3.2 Evaluating neural variability.** We first measure the temporal neural variability of the latent state activity at stimulus onset for MCPC and PC. This mimics the neural variability recording in the V1 region of cats done by Churchland et al. [65]. The model used for this experiment is trained on MNIST with MCPC to optimize the model's FID. Moreover, we measure the neural variability around 256 stimuli onsets from the MNIST evaluation set. We measure the temporal neural variability by computing the standard deviation of the activity of all the latent states over a sliding window of 1000 timesteps. Then, we average the neural variability over all latent states for the 256 stimuli onsets. Churchland et al. [65] employed a 50-ms sliding window to estimate the variance in membrane potential of individual neurons. Then, they averaged the neural variability across the 52 recorded neurons and all stimuli to plot the mean change in neural variability at stimulus onset. Our approach replicates the experimental approach of Churchland et al. [65] for measuring neural variability of membrane potentials from cat V1.

**4.3.2 Similarity of spontaneous and average evoked neural activity.** Our method to measure the similarity of spontaneous and average evoked activity follows the approach used

to measure this similarity in the V1 region of ferrets. Berkes et al. [38] recorded the spontaneous and average evoked neural activity with a linear array of 16 electrodes implanted in V1 of 16 ferrets (approximately 4 ferrets per age group). They measured the evoked activity for natural stimuli, sinusoidal gratings, and random noise. Moreover, they quantified the similarity in activity using the KL divergence. Our approach relies on similar sensory inputs and uses the same quantification metric. We measure the similarity between the spontaneous activity and the average evoked activity using a KL divergence in 10 MCPC models. We compute the KL divergence at different steps during training on MNIST as follows: First, we record the evoked activity of an MCPC model to (i) 256 samples from MNIST's evaluation set (natural stimuli), (ii) 256 samples of sinusoidal gratings with 16 possible orientations, and (iii) 256 samples of random binary images. We record from five randomly selected latent states in the first latent layer ($l$ = 1) for 9,500 inference steps. Recording from five latent states reduces the computation time while maintaining representative results for the whole network. Moreover, recording from the first latent layer mirrors the V1 region which is the first cortical area that processes visual information. Second, we record the spontaneous activity of the model for the same five latent states for 9,500 activity updates. Third, for each type of evoked activity, we compute the average experimental distribution of evoked activities across samples. Finally, we compare the three average distributions to the distribution of spontaneous activity using the KL divergence as implemented by Pérez-Cruz [99]. We repeat this procedure for 10 models trained on MNIST with MCPC to optimize the model's FID. These models have the same architecture and learning parameters but they are initialized using a different seed. After, the KL divergence for natural stimuli is compared using paired samples t-tests to the KL divergence for gratings and for noise.

## Supporting information

**S1 Appendix. Learning in a linear model with one input neuron and one latent state using MCPC and PC.**
(PDF)

**S2 Appendix. Predictive coding optimizes an infinitely loose bound on the marginal log-likelihood ln $p(y; \theta)$.**
(PDF)

**S3 Appendix. Proofs for MCPC capturing the variability of cortical activity.**
(PDF)

**S4 Appendix. Hyperparameter search results.**
(PDF)

**S1 Fig. Masked inference and similarity increase across model hierarchy. top**. We visualize the latent layer activity for a masked input (top left) across all latent layers for PC and MCPC, following the method outlined in Section 4.2.2 of the manuscript. The MCPC model identifies different potential interpretations for a given masked input across its latent layers, whereas the PC model infers only one possible interpretation. Additionally, we visualize the reconstructed images by the MCPC model when the latent layers represent two different possible interpretations. The reconstructed digit when the MCPC model infers a "4" resembles the digit four, while the reconstructed digit when the model infers a "9" resembles the digit nine. **bottom**. We repeat the analysis to assess the similarity between spontaneous activity and average evoked activity across all latent layers, following the method described in Section 4.3.2. The KL divergence between spontaneous activity and average evoked activity for natural stimuli is

lower compared to noise images and image gratings for an MNIST-trained MCPC model across all latent layers. However, this difference is only statistically significant in the first latent layer $x_1$ and not in the higher latent layers.
(PDF)

**S2 Fig. Negative joint log-likelihood of model during MCPC inference.** Sampling using Langevin dynamics has been reported to exhibit large mixing times that scale exponentially with the number of dimensions. We employed at least 50 MCPC inference steps along with a PC warmup before sampling for all our experiments. This figure illustrates the negative joint log-likelihood of MCPC models, $F = \frac{1}{2} \sum_{l=0}^{L-1} \frac{\|x_l - W_l \cdot f(x_{l+1})\|^2}{\sigma^2} + \frac{1}{2} \frac{\|x_L - \mu\|^2}{\sigma^2}$, during inference averaged for 256 data samples for a model for the Gaussian task $\{W_0 = 2, \mu = 0\}$ (**left**) and of a model trained on MNIST (**right**). This figure demonstrates that the average sum of prediction errors of the model converges in fewer than 50 inference steps, indicating that the models have likely reached a steady state. This result suggests that MCPC's convergence time remains manageable even as the size of the latent state increases from one neuron in the Gaussian task to 276 neurons across three layers for the MNIST task. However, larger model might require a convergence time that is beyond practical limits. All the latent variables are randomly initialised before inference following a uniform distribution between -10 and 10. Both models have a learning rate of 0.1 for the activity updates. The shaded region represents the interquartile range.
(PDF)

**S3 Fig. MCPC with and without PC warm-up inference steps trained on the MNIST dataset.** We train an MCPC model on the MNIST dataset with and without warm-up steps. Moreover, we evaluate a range of inference step counts. When the model is trained without warm-up steps, the inference process includes MCPC mixing steps followed by a single MCPC sampling step. However, when the model undergoes training with PC warm-up steps, the initial half of the inference steps consist of PC inference steps, while the remaining inference steps are MCPC mixing steps and one MCPC sampling step. The model parameters for training can be found in S4 Appendix and correspond to the parameters that maximise the FID measure. This figure demonstrates that using PC warm-up inference steps results in improved performance with a limited number of total inference steps, diminished performance with 100 or 200 inference steps, and comparable performance with a large number of inference steps. Ultimately, this result shows that warm-up steps are not always beneficial and should be considered for each learning task separately. The results are shown for three initialisation seeds. The dots show the mean result while the error bars show the standard deviation.
(PDF)

**S1 Video. MCPC posterior inference in linear model.** Animation of the activity of the latent state in a linear model with one latent state during MCPC inference for a constant input. In this animation, the orange dot shows the time-varying activity of the latent state. The blue histogram summarises the activity of the latent state from the beginning of the animation to the time point in the animation being visualized. Finally, the black curve shows the true posterior distribution that can be analytically calculated from the model parameters and the input to the model.
(GIF)

**S2 Video. MCPC posterior inference in non-linear model for half masked MNIST digit.** Animation of the activity of the latent layer $x_L$ in a non-linear model trained on the MNIST dataset during MCPC inference for a half-masked digit input. The orange dot shows the time-

varying activity of the latent state $x_L$ transformed to coordinates using a linear classifier and a convex combination of 10 evenly spaced points on a unit circle. The linear classifier is trained to decode digit class distributions from the latent state $x_L$. The decoded class distribution can then be transformed to a coordinate using the convex combination. The blue hexagons show the probability density of a two-dimensional histogram of the activity of the latent state from the beginning of the animation to the time point in the animation being visualized.
(GIF)

**S3 Video. MCPC posterior inference in non-linear model for full MNIST digit.** Animation of the activity of the latent layer $x_L$ in a non-linear model trained on the MNIST dataset during MCPC inference for a full digit input. The orange dot and the blue hexagons have been determined as described in S2 Video.
(GIF)

**S4 Video. MCPC unclamped activity of sensory input neuron in a linear model.** Animation of the activity of the input neuron in a linear model with one latent state and one input neuron resulting from MCPC dynamics when the input neuron is unclamped. The orange dot shows the input neuron activity over time. The histogram summarises the activity of the input state from the beginning of the animation to the time point in the animation being visualized. The black curve shows the marginal likelihood that can be analytically calculated from the model parameters.
(GIF)

**S5 Video. MCPC unclamped activity of sensory input neurons in non-linear model trained on MNIST.** Animation of the activity of the input neurons in a non-linear model trained on MNIST resulting from MCPC dynamics when the input neurons are unclamped.
(GIF)

## Acknowledgments

We thank Mate Lengyel for comments on an earlier version of the manuscript.

## Author Contributions

**Conceptualization:** Alexander Meulemans.

**Data curation:** Gaspard Oliviers.

**Formal analysis:** Gaspard Oliviers.

**Investigation:** Gaspard Oliviers.

**Methodology:** Gaspard Oliviers, Alexander Meulemans.

**Software:** Gaspard Oliviers.

**Supervision:** Rafal Bogacz, Alexander Meulemans.

**Visualization:** Gaspard Oliviers.

**Writing – original draft:** Gaspard Oliviers.

**Writing – review & editing:** Rafal Bogacz, Alexander Meulemans.

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
