## [Decision Letter · Decision Letter 0]

13 May 2024

Dear Mr Oliviers,

Thank you very much for submitting your manuscript "Learning probability distributions of sensory inputs with Monte Carlo Predictive Coding" for consideration at PLOS Computational Biology.

As with all papers reviewed by the journal, your manuscript was reviewed by members of the editorial board and by several independent reviewers. In light of the reviews (below this email), we would like to invite the resubmission of a significantly-revised version that takes into account the reviewers' comments.

We cannot make any decision about publication until we have seen the revised manuscript and your response to the reviewers' comments. Your revised manuscript is also likely to be sent to reviewers for further evaluation.

Sincerely,

Boris S. Gutkin

Academic Editor

PLOS Computational Biology

Daniele Marinazzo

Section Editor

PLOS Computational Biology

Reviewer's Responses to Questions

**Comments to the Authors:**

Reviewer #1: I enjoyed reading your account of predictive coding, in which variational updates are replaced with (MCMC) sampling. I also appreciate your attempt to establish the validity of the scheme in relation to neuronal responses to stimuli. In many respects, this was compelling work. However, I do not think this report can be published in its current form. This is because you have wandered too far away from conventional formulations of predictive coding, in two senses:

First, your understanding and portrayal of predictive coding is colloquial. You are effectively working in a (machine learning) bubble and are discovering well-known aspects of predictive coding. I suspect part of the problem here is that you are not familiar the predictive coding literature beyond machine learning. As such, you cannot appeal to existing formulations, especially in relation to variational filtering and its implementation in the brain.

The second issue is your strange choice to:

"encode layer variance in the noise variable to make the dependence between the level of noise in MCPC’s dynamics and the variance of its generative layers explicit."

This choice is understandable if you thought that it would make it easier for you to reproduce empirical (neurophysiological) responses. However, by failing to incorporate the noise variance (i.e., precision) in your objective function your F is not a “negative joint log likelihood”. Your choice is odd given your previous work showing how the noise variance or precision can be itself optimised with respect to F (i.e., variational free energy) (Bogacz, 2017).

In summary, I think you need to rewrite your paper starting with a clear definition of what you understand by predictive coding. Predictive coding was introduced in the 1950s for compressing sound files (Elias, 1955) and was first proposed in neuroscience for retinal processing (Srinivasan et al., 1982). Related proposals for hierarchical predictive coding were then discussed in terms of cortical computations and particle filtering (Lee and Mumford, 2003; Mumford, 1992). Hierarchical predictive coding was foregrounded by Rajesh Rao and Dana Ballard (Rao and Ballard, 1999) and subsequently shown to be equivalent to extended Kalman filtering (Friston and Kiebel, 2009; Rao, 1999), which itself is an instance of variational filtering (Friston, 2008b). I mention variational filtering because I think you need to explain how your formulation (e.g., equation 3) differs from the second lemma in (Friston, 2008b). (E.g., equation 12).

The answer is that you are considering a limited case in which you have ignored dynamics. In generic predictive coding schemes, one normally works in generalised coordinates of motion. The Kalman filter is an example of just dealing with first-order motion. Your so-called PC is a zero-order approach in which you ignore motion altogether, so that you can focus on static image classification problems.

You need to make it clear that your predictive coding is not predictive coding in the general sense. I think you need to find an appropriate acronym; perhaps MLPC for maximum likelihood or machine learning predictive coding?

If you follow through the work on variational filtering you should appreciate that putting the noise variance into the objective function still allows you to link your formulation to stimulus dependent fluctuations in noise levels. This is because when a stimulus arrives the precision weighted prediction errors mean that the curvature of the free energy increases and the excursions around the Langevin flow are attenuated. In other words, the phenomenology you are trying to demonstrate is an emergent property of variational filtering. Put another way, when noise variance increases the curvature of F increases and neurons spread out or diffuse further. This means that if the random fluctuations on motion have unit variance, they will appear to have greater variability when there is no stimulus or there is a loss of precision (by Weber’s law). Interestingly, the sample density of neurons – with unit state noise – approximate the posterior density.

I suggest that you test this out using numerical studies. It will be important to do this because the literature on predictive processing emphasises the minimisation of precision weighted prediction errors: e.g., (Ainley et al., 2016; Clark, 2013; FitzGerald et al., 2015; Haarsma et al., 2018; Kanai et al., 2015; Kok et al., 2012; Limanowski, 2022; Shipp, 2016). This is sometimes cast in terms of estimating the Kalman gain in terms of things like the hierarchical Gaussian filter: e.g., (Palmer et al., 2019).

Minor comments

Could you nuance your abstract and make it clear that Monte Carlo predictive coding is not a “unification of individual frameworks”. Combining predictive coding objective functions with sampling is the basis of particle and variational filtering. I recommend you read about particle filtering and its proposals as a metaphor for stochastic neuronal dynamics in brain hierarchies (Dauwels, 2007; Lee and Mumford, 2003). If you pursue this line of thinking, one can then see that predictive coding without sampling (i.e., viewed as a variational scheme) is simply a description of density dynamics (e.g., variational filtering) under idealised assumptions about sampling. You can find a detailed discussion of this in the companion paper to the variational filtering paper that frames things in terms of dynamic expectation maximisation (Friston et al., 2008). The link between sampling and variational schemes is sometimes expressed as follows: the brain can either be described as performing approximate Bayesian inference exactly (using variational updates) or as performing exact Bayesian inference approximately (using sampling schemes). The former is just a description of the density dynamics you would see under the latter.

On page 2, please say "for understanding how the brain infers and learns."

On page 3, please say "by learning efficient generative models, the brain…"

“Accuracy” is not the objective. The objective is to maximise the marginal likelihood and ensure an efficient and generalisable model.

At the bottom page 3, please remove assertions that models implementing the Bayesian brain principle do not adhere to your three constraints. There are numerous examples that contradict this assertion. You can find a review in (Friston, 2008a)

On page 4, please remove any intimation that predictive coding "only infers the most likely state of the environment from sensory inputs". It may be that your implementations were restricted in this way. However, generalised predictive coding schemes estimate a full posterior over both states and parameters. (Please see below).

On page 18, you are correct to say that the expectation maximisation scheme is equivalent to a Dirac delta function over the true posterior: but not over the latent states, rather over model parameters. The notion that sampling "optimises a tightly bound on the log p(y ; theta) (proposition 3) is not an explanation for the failures of expectation maximisation. The failures of variational expectation maximisation relate to the fact that it ignores uncertainty about the parameters. In other words, in contrast to generic predictive coding schemes, machine learning implementations assume a point mass over the parameters and are therefore unable to evaluate the marginal likelihood that needs to be optimised; i.e., p(y). This is a subtle but fundamental issue which confounds much of machine learning — but is resolved in predictive coding.

On page 23, you say that “the scalar variance of the noise is not specified in the requirements the fluctuation dissipation theorem." However, this does not mean that "it can be arbitrarily assigned." The noise variance has to match the precision of the data and needs to be estimated in predictive coding. As noted above, your group have done some excellent work along these lines showing how precisions can be estimated in a biologically plausible fashion. I think you need to make this clear and do not leave the reader with the impression that Sigma squared can somehow be assigned arbitrarily.

Please remove the assertion on page 27 "Despite these results, PC has encountered challenges in explaining dynamic features of cortical activity”. There is a large literature addressing the biological plausibility of predictive coding: e.g. (Shipp, 2016; Walsh et al., 2020) . If you are referring to your own work, then you need to change the acronym PC. It may be that you are referring to your static implementation which, by definition, cannot explain "dynamic features of cortical activity." If you wanted to refer to future challenges for your formulation, the usual testbed in neurobiology would be mismatch negativity and oddball paradigms, where one introduces prediction errors by experimental design.

I hope these comments help should any revision be required.

Ainley, V., et al., 2016. 'Bodily precision': a predictive coding account of individual differences in interoceptive accuracy. Philos Trans R Soc Lond B Biol Sci. 371.

Bogacz, R., 2017. A tutorial on the free-energy framework for modelling perception and learning. J Math Psychol. 76, 198-211.

Clark, A., 2013. The many faces of precision (Replies to commentaries on "Whatever next? Neural prediction, situated agents, and the future of cognitive science"). Front Psychol. 4, 270.

Dauwels, J., 2007. On Variational Message Passing on Factor Graphs. In: 2007 IEEE International Symposium on Information Theory. Vol., ed.^eds., pp. 2546-2550.

Elias, P., 1955. Predictive coding–I. IRE Transactions on Information Theory. 1, 16–24.

FitzGerald, T.H.B., et al., 2015. Precision and neuronal dynamics in the human posterior parietal cortex during evidence accumulation. Neuroimage. 107, 219-228.

Friston, K., 2008a. Hierarchical models in the brain. PLoS Comput Biol. 4, e1000211.

Friston, K., Kiebel, S., 2009. Predictive coding under the free-energy principle. Philos Trans R Soc Lond B Biol Sci. 364, 1211-21.

Friston, K.J., 2008b. Variational filtering. Neuroimage. 41, 747-66.

Friston, K.J., Trujillo-Barreto, N., Daunizeau, J., 2008. DEM: a variational treatment of dynamic systems. Neuroimage. 41, 849-85.

Haarsma, J., et al., 2018. Precision weighting of cortical unsigned prediction errors is mediated by dopamine and benefits learning. bioRxiv.

Kanai, R., et al., 2015. Cerebral hierarchies: predictive processing, precision and the pulvinar. Philos Trans R Soc Lond B Biol Sci. 370, 20140169.

Kok, P., et al., 2012. Attention reverses the effect of prediction in silencing sensory signals. Cereb Cortex. 22, 2197-206.

Lee, T.S., Mumford, D., 2003. Hierarchical Bayesian inference in the visual cortex. J Opt Soc Am A Opt Image Sci Vis. 20, 1434-48.

Limanowski, J., 2022. Precision control for a flexible body representation. Neurosci Biobehav Rev. 134, 104401.

Mumford, D., 1992. On the computational architecture of the neocortex. II. The role of cortico-cortical loops. Biol Cybern. 66, 241-51.

Palmer, C.E., et al., 2019. Sensorimotor beta power reflects the precision-weighting afforded to sensory prediction errors. Neuroimage. 200, 59-71.

Rao, R.P., 1999. An optimal estimation approach to visual perception and learning. Vision Res. 39, 1963-89.

Rao, R.P.N., Ballard, D.H., 1999. Predictive coding in the visual cortex: a functional interpretation of some extra-classical receptive-field effects. Nature Neuroscience. 2, 79-87.

Shipp, S., 2016. Neural Elements for Predictive Coding. Front Psychol. 7, 1792.

Srinivasan, M.V., Laughlin, S.B., Dubs, A., 1982. Predictive coding: a fresh view of inhibition in the retina. Proc R Soc Lond B Biol Sci. 216, 427-59.

Walsh, K.S., et al., 2020. Evaluating the neurophysiological evidence for predictive processing as a model of perception. Ann N Y Acad Sci. 1464, 242-268.

Reviewer #2: In this article, the authors present a novel approach to biologically plausible inference of generative models (MCPC). Predictive coding (PC) stands as a dominant theory, wherein information about a signal is reduced to a prediction error. Various implementations of predictive coding have been proposed. The authors offer their own, based on Monte Carlo sampling (MC). The authors compare its performance with the implementation of free-energy PC with delta-function. The authors conduct theoretical tests for the simple linear gaussian model and the digit classification model. The authors show the model's ability to reproduce some properties of neural dynamics in the visual cortex that have not been reproduced by previous PC implementations. The article is clearly written, relevant to the journal's scope, and presents a coherence of implementation and validation procedures. I highly recommend this work for publication, with some revisions as outlined below.

Major issues:

1. The authors somewhat ambiguously introduce the concept of PC, which can be misleading. PC is a general idea that can be implemented in several ways. One of these approaches is free energy, which the authors refer to. It employs variational inference (VI) to approximate the posterior distribution. The difference between MCPC and free-energy PC is that the authors utilize the Monte Carlo approach instead of VI. I believe the authors should carefully describe the conceptual differences of their idea from existing approaches (e.g., Rao & Ballard, PC/BC-DIM, and free energy).

2. The authors claim that PC cannot infer posterior probabilities. It would be more correct to state that this is true for the implementation of PC within the free-energy framework, where the variational distribution is represented as a delta function. The authors should note that the free-energy framework is not limited to this type of distribution. Moreover, the original implementation employs Laplacian approximation, which is integrated, for example, in the SPM library. In the meantime, I agree that in free-energy-PC there won't be a true posterior distribution for complex models. Unlike free-energy-PC, MCPC does not restrict the type of distribution but requires infinite-time Langevin dynamics simulation. Hypothetically, free-energy-PC could use more complex variational distributions. I believe, such comparisons would help to convey the idea of authors.

3. MCMC is famous for its convergence time strictly depending on the dimensionality of the parameter space and the hidden states of the generative model. It would be beneficial to address this issue in the article, showing relevant plots.

Minor issues:

1. The text doesn't clarify how the model for digit recognition was trained. How do you form the batch? How does the batch participate in the MCPC algorithm? Is it true that each gradient step is averaged over all images, for which Langevin sampling occurs separately and independently?

2. The authors mention the potential implementation of a dynamic version (inference of dynamic inputs), but they don't provide details. Since I am skeptical about this possibility, I would like to ask the authors to provide some details regarding this issue.

3. In line 286, what does "locally optimal" mean?

Reviewer #3: In this paper, the authors presented Monte Carlo Predictive Coding (MCPC), a cortical model for probabilistic inference and learning that combines hierarchical predictive coding and Langevin Monte Carlo sampling. The authors demonstrated the validity of the model through two datasets: a simulated one-dimensional Gaussian dataset, and a more complicated MNIST dataset. The results have shown that MCPC can (1) infer posteriors that contain many modes (MNIST) and learn accurate generative models, (2) explain experimental results such as neural variabilities diminishing at stimulus onset, and increasingly similar spontaneous activities to those during natural image perception through development, (3) be robust to different noise levels of the samplers.

I find this paper very well written with quite rigorous descriptions of the model. I have no major concerns on the model/theory itself, and agree with the authors that investigating how sampling theories reconcile with predictive mechanism in the brain is extremely promising. However, I find the scope of the presented results to be restricted and insufficient in convincing me of the novelty of the methods. I will discuss my general concerns first, followed by minor issues.

General concerns:

(1) My biggest worry is that most of the results presented here are, respectfully, not entirely surprising. As the authors noted, the architecture / assumed generative model of MCPC is similar to many previous lines of works that study neural circuits for sampling, especially for one-layer Gaussian generative models. The learning aspect of the model is indeed not common in current literature (especially driven by prediction errors). However, it is not unreasonable to expect EM algorithms would work well in these settings, using Monte Carlo samples. The biological plausibility aspect of the model also mostly inherits predictive coding (local learning rules, error neurons, etc.). Therefore, though I fully believe in the validity of Figure 2-4 presented by the authors, they do not convince me with extra insights provided to neuroscience audiences about the neural circuits, more than an implementation of generative model.

(2) Related to the previous point, in my opinion, the biggest difference of MCPC compared to other sampling models is its **hierarchical structure**, but I do not see many results or discussion about this aspect. For example, an important goal in hierarchical predictive coding is to learn more and more abstract representations along its hierarchy through prediction and error (e.g. bars/edges → corners → objects). It would be fascinating to see (i) if MCPC can still learn meaningful hierarchies of abstractions, and (ii) how do sampling dynamics at each level unfold when there are ambiguities? The authors showed in Figure 2d that the last layer’s latents seem to be bi-modal and encode possible digit identities – what do the other layers look like? Are there multiple modes at other levels or just the last level? I’m also curious to see in the case of the Figure 2d, what does the reconstruction of the digit look like? Are they complete digits? For your results in Figure 5, which level’s activities are these? Are there layer-wise differences?

In summary, I would love to see enhanced results and discussions about the hierarchical aspects of the sampling results, which I believe is the most distinct characteristics of MCPC.

Smaller questions and minor issues:

(1) Could the authors explain why the MNIST images were converted to be binary Bernoulli distributions? Would a Gaussian assumption hurt the performance? It seems that there are continuous values (lighter “white” vs. darker “white”) in the samples shown in Figure 3, why would this be the case if the input is assumed to binary?

(2) In 4.1.1 (and Algorithm 1) there is a “warm-up” step that first performs K steps of gradient descent (without noise) before transitioning to Langevin sampling with noise – is this step necessary? I’m curious if this would actually lead to worse sampling performance (especially in high dimension), if one particular mode of the posterior has a large basin.

(3) Equation 2 is missing the top level prior probability x_L

(4) The authors could unify the symbol for Gaussian pdf (it’s N vs. \\mathcal{N} at different places..)

**Have the authors made all data and (if applicable) computational code underlying the findings in their manuscript fully available?**

Reviewer #1: Yes

Reviewer #2: Yes

Reviewer #3: None

PLOS authors have the option to publish the peer review history of their article (what does this mean?). If published, this will include your full peer review and any attached files.

Reviewer #1: No

Reviewer #2: **Yes: **Olesia Dogonasheva

Reviewer #3: No
---

## [Decision Letter · Decision Letter 1]

1 Oct 2024

Dear Mr Oliviers,

We are pleased to inform you that your manuscript 'Learning probability distributions of sensory inputs with Monte Carlo Predictive Coding' has been provisionally accepted for publication in PLOS Computational Biology.

Best regards,

Boris S. Gutkin

Academic Editor

PLOS Computational Biology

Daniele Marinazzo

Section Editor

PLOS Computational Biology

Reviewer's Responses to Questions

**Comments to the Authors:**

Reviewer #1: Many thanks for attending to my previous requests. And congratulations on a convincing piece of work.

Reviewer #3: I would like to thank the authors for their detailed response to my original comments. I enjoyed reading the added figures and results on learning the data variance in weights and the hierarchical representations.

I find no other major issues with the results and I believe the results will be highly of interest to both predictive coding and neural sampling audiences. I have the following small comments on the new results for the authors to consider:

- If in fact data covariances can be learned and encoded into model weights, what implications do this result have on the neural sampling theory? Specifically, if there is a sudden change in data covariance but not the mean (e.g., you're in the same room, day vs. night), do you think there're fast plasticity rules going on that quickly adjust the model weights to adapt to the new data variances? I think exploring the difference between your approach and how traditionally covariance has been handled (e.g. extracted in precision matrices as an explicit part of the objective function, or implicitly by samplers) is very interesting.

- Re the results on hierarchical representation: Initially, I was surprised when I read that there are bimodal effects on all layer neurons, as I was expecting to see this only happen in "higher" layer neurons that extract digit identities. But then this may make sense since different layers have no receptive field size differences. It would be really cool to see if such ambiguity only happens in higher-layer neurons, if your lower-layer model neurons have a smaller RF size than higher layer neurons (similar to Rao & Ballard (1999), essentially learning a pooling of representations).

**Have the authors made all data and (if applicable) computational code underlying the findings in their manuscript fully available?**

Reviewer #1: Yes

Reviewer #3: None

PLOS authors have the option to publish the peer review history of their article (what does this mean?). If published, this will include your full peer review and any attached files.

Reviewer #1: No

Reviewer #3: No

---

## [Editor Report · Acceptance letter]

15 Oct 2024

PCOMPBIOL-D-24-00439R1 

Learning probability distributions of sensory inputs with Monte Carlo Predictive Coding

Dear Dr Oliviers,

I am pleased to inform you that your manuscript has been formally accepted for publication in PLOS Computational Biology. Your manuscript is now with our production department and you will be notified of the publication date in due course.

With kind regards,

Anita Estes
